# Stereo-random oligonucleotides enable efficient recruitment of ADAR in vitro and in vivo

Laura S. Pfeiffer [1,7], Tobias Merkle[1,7], Paul Vogel[2], Inga Jarmoskaite [2], Jonathan M. Geisinger[2], Ngadhnjim Latifi[1], Marco Herrera-Barrera[3], Feijie Zhang[2,4], Lisa Groß[1], Carolin Schlitz[1], Daniel T. Hofacker [1], Clemens Lochmann [1], Davide Fumagalli [1], Stefanie Gackstatter[1], Vanessa Deisling[1], Mark A. Kay [2,4], Jin Billy Li [2] & Thorsten Stafforst [1,5,6] ✉

Site-directed RNA editing is a promising and potentially safer alternative to genome editing. Previous methods have been developed that recruit the endogenously and ubiquitously expressed ADAR enzymes to initiate site-specific A-to-I edits, but often suffer from low efficacy or dependency on viral delivery. Chemically modified oligonucleotides may be a promising alternative, but the approach still lacks systematic in-depth studies. Furthermore, the best characterized platform uses stereo-pure backbone chemistry, which is not widely used, commercially unavailable and challenging to manufacture. Here, we report on single-stranded oligonucleotides of 30-60 nt length, which are fully chemically stabilized by applying commercially available, classical RNA drug modifications, like 2´-O-methyl, 2´-fluoro, and DNA on a stereo-random phosphate/phosphorothioate backbone. We demonstrate our so-called RESTORE 2.0 oligonucleotides to induce the correction of pathogenic point mutations, efficacy after GalNAc-mediated uptake into human primary hepatocytes, and proof of in-vivo efficacy in mice upon lipid nanoparticle-mediated delivery. The discovered design principles may increase the accessibility of site-directed RNA base editing to expand and support further research in this field.

Adenosine-to-inosine (A-to-I) RNA editing is a natural, post-transcriptional RNA diversification mechanism in metazoans[1]. Endogenously, A-to-I editing can alter start or stop codons, recode amino acids, and influence RNA splicing, among other mechanisms[2]. For this reason, RNA editing has recently gained substantial interest for its therapeutic potential to correct pathogenic gene mutations by steering editing activity to any desired site in a specific transcript - an

approach called site-directed RNA base editing (SDRE[3,4]). In contrast to DNA (base) editing approaches, SDRE is reversible, can be started and stopped, tuned in a dose-dependent manner, and raises fewer safety and ethical concerns regarding germline manipulation[4].

Several SDRE approaches have already been described. However, most of them are based on overexpressed, engineered editing enzymes[5–7] that elicit high amounts of global off-target editing[8–11],

[1]Interfaculty Institute of Biochemistry, University of Tübingen, Tübingen, Germany. [2]Department of Genetics, Stanford University, Stanford, CA, USA. [3]Stanford Institute for Stem Cell Biology and Regenerative Medicine, Stanford University, Stanford, CA, USA. [4]Department of Pediatrics, Stanford University, Stanford, CA, USA. [5]Gene and RNA Therapy Center (GRTC), Faculty of Medicine, University of Tübingen, Tübingen, Germany. [6]iFIT Cluster of Excellence (EXC2180) "Image-Guided and Functionally Instructed Tumor Therapies, University of Tübingen, Tübingen, Germany. [7]These authors contributed equally: Laura S. Pfeiffer, Tobias Merkle. ✉e-mail: thorsten.stafforst@uni-tuebingen.de

suffer from delivery or localization challenges[9,12] and may induce other, unknown long-term detrimental consequences[4]. A possible solution to these problems is the use of endogenous editing enzymes for SDRE including the A-to-I editing enzymes adenosine deaminase acting on RNA (ADAR)[13–18]. Two catalytically active ADARs are known in humans: ADAR1 and ADAR2. Both ADARs have a C-terminal deaminase domain (DD) that facilitates the deamination reaction, and two (ADAR2) or three (ADAR1) N-terminal double-strand RNA binding domains (dsRBDs)[2]. Additionally, ADAR1 has two ubiquitously expressed isoforms: the N-terminally truncated, constitutively expressed p110 isoform and the p150 isoform, which is expressed under the control of an interferon-inducible promotor[19,20]. By contrast, ADAR2 expression is mostly limited to the brain and artery[21]. All ADARs are localized in the nucleus[22,23], although the ADAR1p150 isoform can also shuttle to the cytoplasm due to an N-terminal nuclear export signal (NES)[24].

While a full-length structure of ADAR has not been achieved yet, structural data based on ADAR2 is available that supported the development of SDRE methods[25–27]. Earlier structural data of the ADAR2 DD bound to a dsRNA substrate shows a base-flipping mechanism underlies the adenosine deamination process[25]. This structure also discloses a plethora of ADAR2 DD interactions around the edited adenosine and the orphan cytidine. A more recent structure shows substrate binding of an asymmetric homodimer of truncated ADAR2, which is missing one of its two dsRBDs[26]. In this structure, the catalytically active DD at the edited adenosine is sandwiched between the DD of the other monomer and the dsRNA substrate, while the dsRBD of the inactive ADAR2 monomer engages with the dsRNA substrate in the 3′-direction of the target transcript. A respective footprinting assay showed that the minimum antisense ON length required by an asymmetric ADAR2 homodimer is 15 nt 3′-adjacent and 26 nt 5′-adjacent to the orphan cytidine base, resulting in a total antisense length of approximately 42 nt if only a single dsRBD of each ADAR2 monomer is engaged in dsRNA binding[26]. Furthermore, high-resolution structural information proved valuable to design editing-enhancing non-canonical nucleobase modifications such as the Benner's Base Z or nebularine as substitutes for the orphan cytidine base in the ON[28]. Structural information supported the finding that ribose in locked confirmation, e.g., LNA (locked nucleic acid), is only accepted in specific positions relative to the on-target adenosine[28,29].

Most SDRE tools that use endogenous ADAR are based on the viral delivery of genetically encoded guide RNAs (gRNAs)[14–17]. A major challenge of these approaches is the need for very high gRNA expression, which typically requires strong polymerase III promotors in combination with cyclization of the gRNA. Additionally, precise, bystander-free editing with high on-target efficiency remains challenging. Finally, limited expression strength and tropism of the viral vehicle may currently restrict its application to certain tissues and redosing might be difficult due to host immunity against the virus vehicle[30]. In this context, an attractive alternative is the use of chemically modified oligonucleotides (ON) that recruit endogenous ADARs[13,18,28]. Several drug modalities based on chemically modified ON, e.g., siRNAs and ASOs, have been clinically approved recently[31], confirming that ON approaches are feasible in principle. Even though there is enormous industrial interest in the field[32–34], systematic publications reporting design rules of chemically modified ON that recruit endogenous ADAR efficiently remain scarce, particularly designs containing only commercially available and readily accessible modifications.

Our previously published approach[13], termed RESTORE (1.0), used 95 nt long oligonucleotides (ON) comprised of a 40 nt long specificity domain (mediating programmable binding to the target mRNA) plus a 55 nt structured ADAR-recruiting domain. Although applicable and functional, it suffered from severe limitations. First, with ~95 nt the overall size was too large, limiting ON manufacturing as well as delivery. Second, editing efficiency was moderate and partly dependent on

the induction of the ADAR1p150 isoform via interferon-α induction. Finally, the RESTORE 1.0 ON was only partly chemically modified and suffered from low metabolic stability.

Today, the best characterized and functional oligonucleotides are AIMers. AIMers are approximately 30 nt long ON containing a 15 nt stretch of 2′-F modifications at its 5′-terminus and a 2′-OMe modification block at its 3′-terminus, which is interrupted by a DNA-modified central base triplet (CBT)[18]. Notably, these ON use stereo-pure phosphorothioate (PS) and phosphoryl guanidine (PN) internucleoside linkages. While these designs perform very efficiently, their high activity was shown to be reliant on the use of the stereo-pure PS and PN linkages. However, whether there is a real benefit to using stereo-pure linkages for RNA therapeutics is highly debated[31,35–37]. Many studies claim improved potency and performance of stereo-pure ONs compared to the respective stereo-random ON design[38–40]. By contrast, a large side-by-side in vitro and in vivo study on stereo-pure vs. stereo-random gapmers showed no improvement of potency or performance, and that the sequence and type of chemical modifications had a much larger impact on the respective performance and therapeutic profile[37]. Additionally, stereo-pure linkage chemistry is commercially not accessible, difficult to manufacture and thus limiting its use in academia and industry.

Here, we describe design rules for the development of fully chemically stabilized and short (30-60 nt) ON, which enable the potent recruitment of endogenous ADAR for efficient and precise RNA base editing in cell lines, primary cells, and in vivo. Importantly, our so-called RESTORE 2.0 ON use a mixed stereo-random phosphate/phosphorothioate (PO/PS) backbone combined with commercially available and clinically used ribose modifications, such as 2′-o-methyl (2′-OMe), and 2′-F ribonucleotides, breaking ground for broader academic and commercial drug research.

## Results

### Basic design principles of unstructured ADAR-recruiting ON

Previous ON designs using stereo-pure backbone modifications have been reported to be 30 nt long[18]. Since the stereo-pure backbone modifications may influence how much the ON can be shortened, we first tested what stereo-random ON length is necessary to efficiently recruit ADAR, using our previously published 95 nt RESTORE 1.0 ON[13] as a reference. The structured, 55 nt ADAR-recruiting domain could theoretically be replaced by simple extension of the specificity domain from a dsRNA sufficiently sized for ADAR binding (Fig. 1A). A recent crystal structure analysis and footprint of the dimer of a truncated ADAR (deaminase domain + one dsRBD) suggested that a substrate of 40−45 bp would also be well suited for ADAR binding[26]. Based on this, we selected a 59 nt single-stranded ON targeting a 5′-UAG codon (target adenosine underlined) in the ORF of human GAPDH as a starting point. Interestingly, the symmetric 59 nt ON (v117.1; 5′− 29-1-29, with "1" depicting the orphan cytidine mismatching the target adenosine) already gave remarkably good editing yields on the endogenous GAPDH transcript in HEK293 cells stably overexpressing different ADAR isoforms (Fig. 1B). Since shortening of the symmetric ON resulted in a strong reduction of editing yield, alternative asymmetric ONs were designed in accordance with the asymmetric footprint of ADAR binding[26], where the orphan cytidine base was shifted more towards the 3′-end of the ON. This asymmetric design could be shortened down to 35 nt (5′−24-1-10, v122.1) before a notable drop in editing was observed. The symmetric 59 nt design v117.19 (5′−29-1-29) and the asymmetric 45 nt design v120.2 (5′−34-1-10) raised our interest for a deeper analysis.

Based on the symmetric (5′−29-1-29) and asymmetric (5′−48-1-10) ON design, we tested the recruitment of endogenous ADAR in HeLa cells instead of overexpressed ADAR isoforms in Flp-In cells. We used symmetric 59 nt ON carrying three consecutive 2′-O-methyl (2′-OMe) modifications at the termini to block exonucleolytic degradation

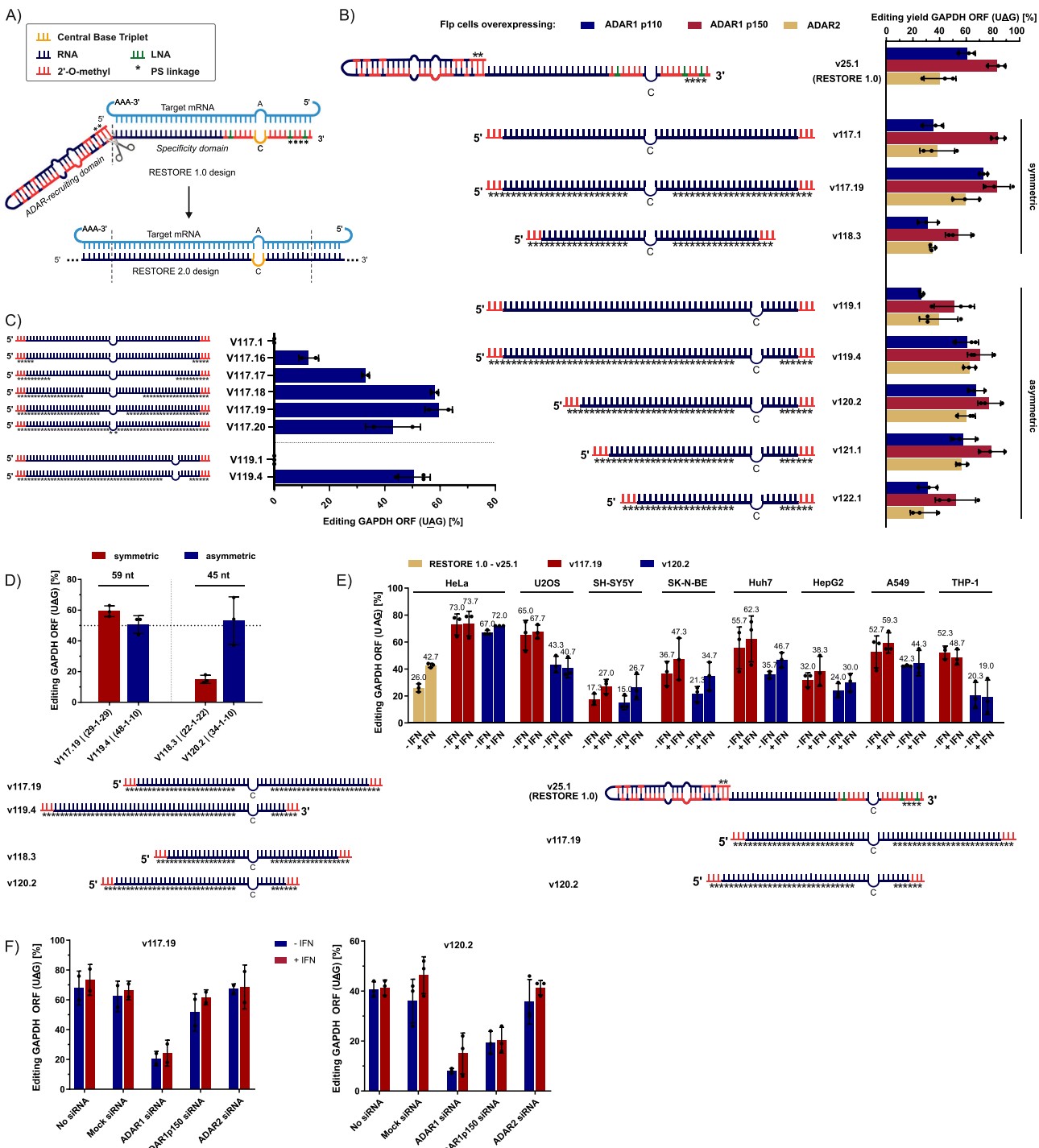

**Fig. 1 | Initial design principles of unstructured ADAR-recruiting ON. A** The ON design applies an extended specificity domain to compensate for the removed ADAR-recruiting hairpin. **B** Editing efficacies of different symmetric and asymmetric ON designs, with and without PS modifications, in Flp-In cells overexpressing the human ADAR isoforms ADAR1p110, ADAR1p150, or ADAR2 (n = 3). **C** Editing yields of ON with increasing PS content that recruit endogenous ADAR in HeLa cells (n = 2 for v117.X, n = 3 for v119.X). **D** Editing yields of different ON symmetries in respect to their lengths, with endogenous ADAR in HeLa cells (n = 3). Short ON require an asymmetric design (longer 5′-terminus) while long ON work in all symmetries equally well.

**E** Editing yields of minimally modified ON lead designs v117.19 and v120.2 in comparison to the RESTORE 1.0 lead, v25.1, in various cell lines (n = 3). All conditions were tested with or without interferon-α (IFN) supplementation. **F** Editing yields of minimally modified ON lead designs v117.19 and v120.2 after knockdown of different ADAR isoforms via RNA interference in HeLa cells (n = 2 for v117.19, n = 3 for v120.2). Data is shown as the mean ± s.d. All editing experiments were performed with 50 nM ON, each datapoint represents a biological replicate. Figure 1A contains an icon created in BioRender (Stafforst, T. (2025), https://BioRender.com/zfhahgx.

(v117.X in Fig. 1C). Editing yields correlated with the phosphorothioate linkage (PS) content, but PS placement directly at the central base triplet (CBT) attenuated editing, identifying a modification-sensitive area. Thus, we kept a symmetric gap of eight unmodified phosphate

linkages around the orphan cytidine (v117.19 and v119.4 in Fig. 1C). Additionally, we confirmed our initial observation that an asymmetric ON design with a longer 5′-terminus was required for shorter oligonucleotides compared to the longer counterparts (Fig. 1D).

The best performing long ON (v117.19) and short ON (v120.2) were further characterized side-by-side for the harnessing of endogenous ADAR in a panel of eight different immortalized cell lines (HeLa, U2OS, SH-SY5Y, SK-N-BE, Huh7, HepG2, A549, THP-1, Fig. 1E) in the presence and absence of IFN-α. The tested ON achieved up to 70% editing, and, in contrast to our previously published RESTORE 1.0 designs[13], were largely independent of IFN-α (which induces ADAR1p150). In some cell lines, where the editing yields were generally lower (most likely due to a higher transfection bias), v117.19 was superior to v120.2. An ADAR-knockdown via RNA interference further confirmed that the long design v117.19 majorly recruited the constitutive ADAR1p110 isoform, and v120.2 seemed to recruit both ADAR1p110 and ADAR1p150 isoforms, although IFN-α had no detectable influence on editing yields (Fig. 1F). This also agrees with what has been reported on the stereo-pure, 30 nt long ONs[18].

## Partial stabilization by pyrimidine modification

Due to the high content of unmodified ribonucleotides, our two lead designs, v117.19 and v120.2, were readily degraded in 100% FBS (Supplementary Fig. 1). Ribose modifications are known to enhance nuclease stability and to improve the pharmacological properties of RNA drugs[41]. For example, FDA-approved siRNAs[42] typically contain 2´-fluoro (2´-F) and 2´-O-methyl (2´-OMe), while RNaseH gapmers[36] apply combinations of 2´-H (DNA) and 2´-O-methoxyethyl (2´-MOE). The previously reported AIMer RNA editing platform[18] uses 2´-F, 2´-OMe and 2´-H modifications, and in our own previous approach, we combined 2´-OMe and LNA[13]. To stabilize the current ON, we focused on the incorporation of clinically used ribose modifications, e.g., 2´-F, 2´-OMe, and 2´-H. We soon started partly modifying the ON after we initially found that repetitive or continuous ribose modifications patterns along the whole ON (including patterns that work for siRNAs) caused a strong decrease of editing performance (Fig. 2A). To stabilize the ON by partial modification, our rational was to block RNase A activity, the major RNase in FBS, which has a clear preference to attack at pyrimidine sites, preferably 3´ to uridine[43]. Thus, we placed 2´ modifications only at the pyrimidines of the ON (v117.19 and 120.2) targeting the ORF of the endogenous GAPDH (5´-UAG), in combination with partial PS linkages and terminal 2´-OMe blocks (Fig. 2B and Supplementary Table 1). Depending on the size and chemical nature of a specific 2´-modification, their introduction at all pyrimidine bases affected editing performance differently. Specifically, 2´-MOE (v120.19) was not accepted, 2´-OMe (v120.18) reduced the editing yield, while 2´-F (v120.17) was accepted with almost no loss in editing efficiency compared to the parent ON (v120.2, Fig. 2B). Importantly, the partial modification resulted in a notable increase of nuclease stability. Both lead ONs with 2´F at their pyrimidines (v120.17 and v117.28) were still detectable after 7 days in 100% FBS), while the parent ON was fully degraded within the first 5 min (Fig. 2C, Supplementary Fig. 1). These two designs were tested in three different primary cells, RPE, NHA and NHBE (Fig. 2D), resulting in satisfying editing yields. However, compared to the unmodified analog, partial 2´-F modification always decreased editing yields to some degree. Still, the partial 2´-F modification was sufficient to elicit up to 40% editing upon gymnotic uptake of ON 117.28 into HeLa cells (Fig. 2E).

Harnessing endogenous ADAR, either with chemically modified ONs[13,18] or with genetically encoded guide RNAs[15,17,44,45], has repeatedly been shown to be highly precise and to induce no global off-target edits nor detectable changes in the endogenous editing homeostasis. We checked this again for the partially modified ON v117.28 against the ORF of GAPDH in NHA cells (primary normal human astrocytes), and applied next-generation RNA sequencing to determine the editing homeostasis. As expected, we did not detect off-target editing, changes in the global editing homeostasis (at >19,000 observed endogenous editing sites) or changes in gene expression (Supplementary Fig. 2). But when we transfected the endblocked, but otherwise unmodified, analog ON v117.19, we detected the induction of interferon-stimulated genes and a shift in the global editing homeostasis towards higher editing levels (Supplementary Fig. 2). Thus, the chemical modification of RESTORE ON is required not only to metabolically stabilize the ON but also to mitigate immunogenicity of the ON.

We then transferred the partial stabilization strategy to a human disease-relevant setting. Specifically, we aimed to repair the SERPINA1 E342K mutation (5´-CAA codon context), a common cause of α−1-antitrypsin (A1AT) deficiency[46], in a HeLa cell model overexpressing the mutated cDNA. Besides Sanger sequencing, we analyzed the total A1AT protein levels from the cell supernatant. Notably, the partially 2´-F modified design (v117.25, 46.7% editing) clearly outperformed the unstable parent ON (v117.19, 15.3% editing, Fig. 2F). Interestingly, we found that around half of the 2´-F-modifications could be replaced by 2´-OMe modifications (v117.26). Reducing the 2´-F content may be desirable given their reported involvement in toxicity in certain cases[47–49]. More importantly, the 2´-OMe/2´-F mixed ON (v117.26) even outperformed the 2´-F modified analog design (v117.25) in both editing yield and total A1AT protein restoration. Conclusively, the results show that partial modification of the ON at pyrimidine nucleosides with 2´-F and 2´-OMe is well tolerated in different sequence contexts and is a simple basic principle to enhance ON nuclease resistance in serum without major losses in editing performance.

## Identification of optimal PS/PO patterns in the central base triplet

To further improve ON stability and performance, we investigated which internucleoside linkage (a to j) around the CBT area (nucleotide positions $N_{+4}$ to $N_{-5}$) is amenable for PS modification (see Fig. 3A). For this, we systematically added PS linkages around the CBT in the partially modified SERPINA1 E342K-targeting ON (v117.26). The inclusion of PS at the linkages 5´ to the orphan cytidine ($N_0$), i.e., linkages a – e, was well tolerated and even boosted editing yields (v117.28 and v117.33). The omission of PS at positions b and c further boosted editing yields up to 80% with only a minor loss of nuclease stability (v117.28 vs. v117.39, see Fig. 3B). On the other hand, PS linkages 3´ of the orphan cytosine were less well tolerated and decreased editing yields (v117.28/v117.33 versus v117.39/v117.34, respectively). We assigned linkages g and h to be majorly responsible for editing yield loss. Interestingly, some of the previously reported stereo-pure AIMer designs show a distinct switch from the $S$ to the $R$ stereoisomer at linkage g[18]. For further use, we selected the PS pattern from v117.39 (PS at positions a, d, e, f and j; PO at b, c, g, h, and i) since it provided the optimal balance between high editing yields and stability in 100% FBS and lysosomes. Importantly, v117.39 was not only more stable, but also much more efficient (75% editing yield) compared to the starting ON v117.26 (40% yield) with the full PO-gap around the CBT. We also observed this improvement in stability and/or efficiency for two other targets (Supplementary Fig. 3A–B).

## Design of fully modified RESTORE 2.0 ON

RNA drugs must overcome multiple barriers in order to reach their site of action[50]. Particularly important is the survival in the endo-lysosomal environment[51]. We tested the stability of our pyrimidine-modified, PS-optimized ON by incubation in lysosomal solution and found that even the ON with 100% PS internucleoside linkage modification was completely degraded within 24 h (Fig. 3B). It was reported that lysosomal nucleases like RNase T1/2 prefer slicing 3´ to purines at purine-pyrimidine interfaces[52], suggesting that full 2´ modification of the ON, including the purine nucleotides, is required to obtain stability in the endo-lysosome. Similarly, the previously published, stereo-pure AIMer designs are fully 2´-modified including all purines[18]. We started testing this based on the high-performing 59 nt PS-optimized design v117.39 that targets the SERPINA1 E342K site. Substitution of some or all

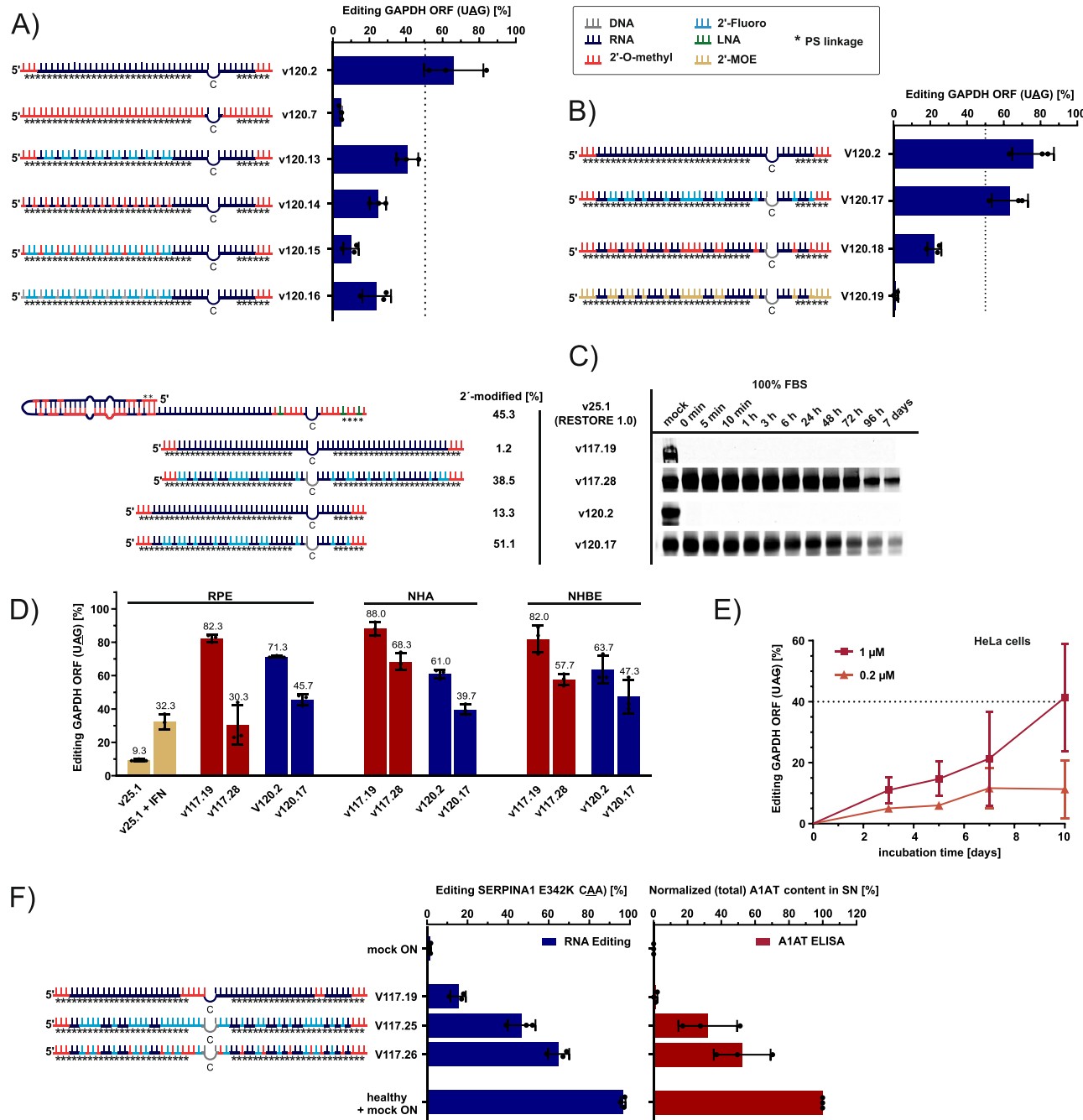

**Fig. 2 | Partial stabilization of the ON. A** Repetitive 2´ modification of large stretches of nucleosides of an ON targeting a GAPDH ORF site. **B** Partial 2´ modification, only at pyrimidine nucleosides. **C** Representative assessment of ON stability with an ON degradation assay in 100% FBS (Comprehensive, uncropped images are in Source data file). **D** Editing performance of partially stabilized ON in RPE, NHA and NHBE cells. GAPDH ORF target. Interferon was only applied with the v25.1 ON. **E** Gymnotic uptake of ON v117.28 (1 μM and 0.2 μM) in HeLa cells. GAPDH ORF target. **F** Repair of the SERPINA1 E342K mutation causing α-1-antitrypsin

(A1AT) deficiency in a HeLa cell model stably overexpressing SERPINA1 E342K cDNA. Editing yields and normalized total A1AT levels in the supernatant of SERPINA1 E342K-piggyBac HeLa cells after treatment with partially modified ON designed to repair the SERPINA1 E342K mutation. The mock ON was the GAPDH ORF-targeting ON v117.19 (see Supplementary Table 1). Data in A, B, D-F is shown as the mean ± s.d., N = 3 independent experiments. Experiments in **A**, **B**, **E** and **F** were done in HeLa cells. All editing experiments were performed with 50 nM ON, each datapoint represents a biological replicate.

remaining purine ribonucleosides with 2´-H (DNA) notably improved lysosomal stability, from <24 h (v117.39) up to >7 days (Fig. 4B and Supplementary Fig. 4A–B). However, full 2´-H modification (v117.42) resulted in a drop of editing performance (Fig. 4A), although it exhibited very high lysosomal stability, lasting over a month (Supplementary Fig. 4C). A similar observation was made when all remaining purine ribonucleosides were replaced by their 2´-F analogs, resulting in a nuclease-stable but editing-deficient ON (v117.62). Interestingly,

modifying the remaining purine nucleotides with a ca. 1:1-mixture of 2´-H and 2´-F resulted in a comparably nuclease-stable, but editing-competent ON. Nearly every tested pattern gave a comparably good performance (v117.59 to v117.77 in Fig. 4A) and was stable in lysosomes (Supplementary Fig. 4C). We speculated that this approximate 2:1:1 ratio of 2´-F, 2´-H, and 2´-OMe resulted in editing-competent ON independent of the nature of the underlying nucleoside (e.g., purine versus pyrimidine) given that larger blocks (e.g., >6 nt) of identical 2´

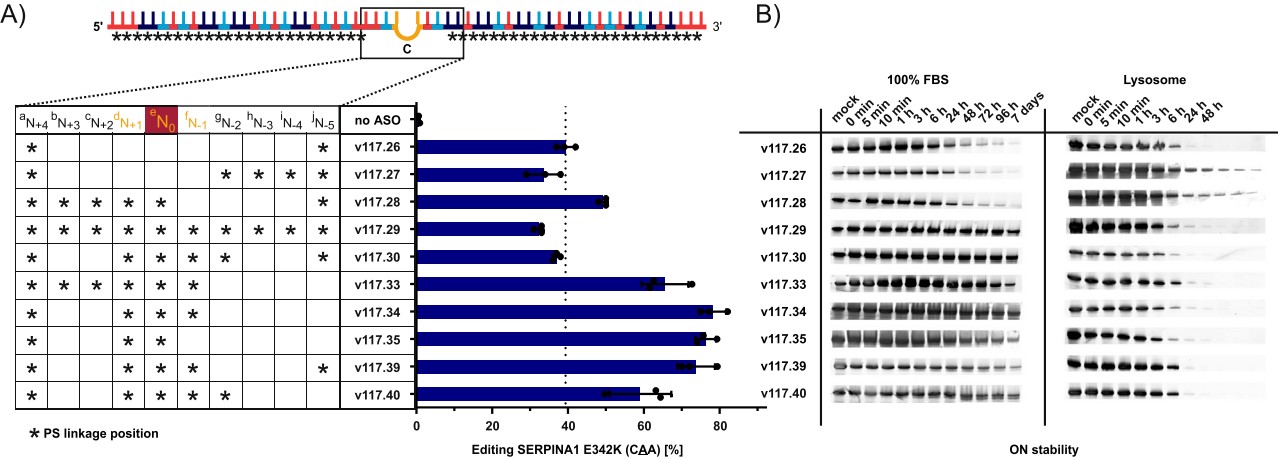

**Fig. 3 | Optimal PS/PO-linkage positioning at the CBT for SERPINA1 a E342K-targeting ON. A** Editing yields upon transfection of 50 nM of different PS/PO combinations in comparison to the former ON design, v117.26. **B** Representative assessment of ON stability with ON degradation assays of the different PS-modified ON in 100% FBS and lysosomal fluid for up to 7 days (Comprehensive, uncropped images are in Source data file). Data in A is shown as the mean ± s.d., N = 3 independent experiments, each datapoint represents a biological replicate.

modifications are avoided. This is in contrast to the previously published, stereo-pure AIMer designs, which use large 2´-F and 2´-OMe blocks[18]. However, our design rule was confirmed by two additional RESTORE ON designs, v117.99 and v117.100, with random placement of said 2´-modification pattern. The fully modified, mixed 2´-F/2´-H design principle was also applicable to 59 nt long ON targeting other transcripts (Supplementary Fig. 4D–G).

Shorter RESTORE ON designs would be desirable, not only to improve manufacturability but also to enhance endosomal escape[53] and to reduce unspecific protein binding, which can sequester the ON or lead to toxicity[54]. Starting from the symmetric 59 nt SERPINA1 ON (5´- 29–1-29, v117.59), we gradually shortened the ON from the 3´-terminus and found only slight reduction of editing yields even after the removal of 19 nt, resulting in a 40 nt ON (5´- 29–1–10, v117.82, Fig. 4C). However, further shortening of the ON down to 31 nt (5´-24–1-6, v117.83) came along with a gradual decrease in editing yield (Fig. 4C). Surprisingly, we discovered that oligonucleotides of two specific short lengths, 32 nt and 33 nt, deviated from this trend and showed unexpectedly high editing efficiency (up to 45%) similar to that of the 40 nt ON (5´-29-1-10, v117.82). We tested several slightly different symmetries of the 32 nt ON (5´-26-1-5 in v117.163, and 25-1-6 in v117.142) and the 33 nt ON (5´-27-1-5 in v117.177, 26-1-6 in v117.141, and 25-1-7 in v117.178), which all performed similar or better than three different symmetries tested for a 38 nt ON. The symmetries of the 33 nt and 32 nt ON overlap very well with the binding footprint of the ADAR dimer involving a single dsRBD of one ADAR monomer[26] and come in very close range of the previously published, stereo-pure AIMer designs (with approx. 30 nt)[18]. Together with the specific 2´-modification and internucleoside modification pattern, we speculate that these short ON symmetries form ideal ADAR substrates with the target RNA, while longer ON, e.g., the 38 nt ON, might enable alternative, editing-incompetent binding registers of ADAR. The folding of an ON into an autoinhibitory secondary structure could also explain why a specific ON sequence is less efficient in editing. However, we could not find any indications for this when we compared the predicted secondary structures of the 40 nt lead ON with the higher performing, shortened version of 33 nt or with the lower performing variants of 34 – 36 nt (Supplementary Fig. 5).

A closer look at the degradation patterns of the fully modified long (59 nt, v117.59) and short (40 nt, v117.82) lead ONs identified key cleavage positions where future engineering might help to improve the performance of the ONs, see Supplementary Fig. 6. In 100% FBS, the DNA gap and the 3 nt 3´-O-methylated uridine tract were the main

cleavage sites, which agrees with the presence and specificity of DNase I and RNase A in FBS. In lysosomes, we identified three key slicing sites in the 3´-half of the long ON. Following RNaseT´s specificity, an adenosine was followed by a pyrimidine base at those three sites. As all three sites were no longer contained in the short lead ON v117.82, no clear cleavage pattern in the lysosome was detected.

For the 59 nt ON, the composition of 2´-F versus 2´-H had strong influence on editing efficiency, strongly favoring a 1:1 mixture of these 2´-modifications (Fig. 4D). We revisited this for the shortened ON. Specifically, we tested 100% 2´-F versus 100% 2´-H versus a 1:1 mix at the remaining purine nucleosides on the 40 nt ON design (5´-29-1-10, v117.82). Unexpectedly, we found that the 100% 2´-F version (117.123) achieved similar editing levels as the 1:1 mixmer (v117.82), while the 100% 2´-H version (v117.93) resulted in very low editing levels (Fig. 4D). For the even shorter ON, e.g., 32 nt and 33 nt, we found that the 100% 2´-F ON (e.g., v117.168, v117.170) clearly outperformed the 1:1 mixmer ON of the same length and symmetry (v117.141, v117.142), further increasing editing yield of the 32 nt ON to ca. 50% and of the 33 nt ON to ca. 60%. These findings suggest that the ON length influences the 2´-modification preferences.

In order to recruit (nuclear) ADAR1p110, the ON must be localized in the nucleoplasm. Compared to our prior RESTORE design[13], this might be facilitated by the reduced length and/or by the increased PS content of the current ON. To address this, we tested two fully modified 40 nt ON with varying PS content targeting the human SERPINA1 E342K site. ON v117.137 had the optimized, high PS content (34/39 linkages, 87% PS), whereas ON v117.109 had a minimal stabilization around the more labile DNA nucleotides with a PS content of 19/39 linkages (49%), see Supplementary Fig 7A. For fluorescence microscopy, Atto488-labeled ONs were spiked into ON transfections in HeLa cells, with an increasing total amount of ON, ranging from 7.5 nM to 50 nM. Both ON clearly localized to the nucleus (Supplementary Fig 7B). However, different to previous studies on RNaseH-recruiting, highly hydrophobic DNA/PS gapmers, both RESTORE ONs rarely formed nuclear aggregates under the used conditions[55]. We applied our previously established assay to determine the intracellular protein interactions of the ON via proximity biotinylation[56]. The detected binders covered various known DNA/RNA-binding proteins, including many that are localized in the nucleus, like SFPQ and NONO, but also LRPPRC (Supplementary Fig 7C). The PS-rich ON tended to identify some of those binders with slightly higher enrichment, indicating a stronger interaction[55,57]. In the context of gapmers, the combination 2´F and PS linkage sometimes induced toxicity, which could be linked

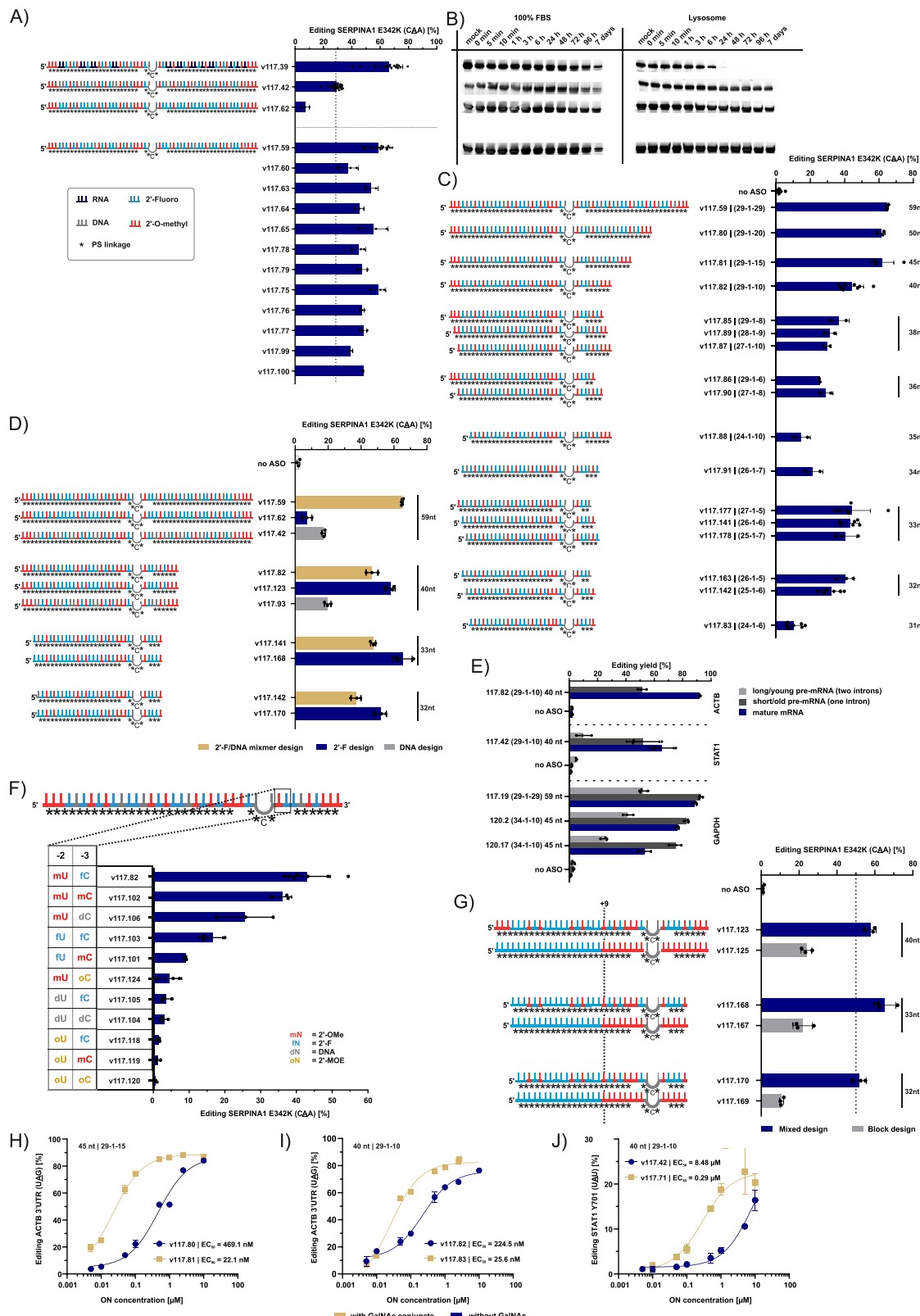

to the degradation of NONO by direct gapmer-binding[58]. However, we could not detect an effect of RESTORE ONs on NONO stability (Supplementary Fig. 7D).

To further check if editing with the ON had happened in the nucleus, we determined the editing yields of several different ONs on three different targets (ACTB, STAT1, GAPDH) on the pre-mRNA versus the mature mRNA level (Fig. 4E). For the fully stabilized 40 nt ON (117.82)

against ACTB, we found editing yields of 93% (mature) and 52% (pre-mRNA), clearly indicating that editing has taken place at least in a large fraction in the nucleus. The splicing kinetics differs between introns in an mRNA, thus the editing yield of a specific pre-mRNA species may often not reflect the editing yield of the fully spliced mRNA prior to nuclear export, meaning that the ACTB mRNA might be fully edited inside the nucleus, even though a yield of only 52% was observed for the indicated

**Fig. 4 | Fully modified RESTORE 2.0 ON. A** Editing performance of fully versus partially modified SERPINA1 E342K-targeting ON with endogenous ADAR in HeLa cells stably expressing the cDNA of SERPINA1 with E342K mutation ($n = 2–17$). **B** Representative assessment of ON stability with a side-by-side stability assay of partially (v117.26, v117.39) vs. fully modified SERPINA1 ON (v117.42, v117.62, v117.59). Uncropped gel images can be found in Source data file. **C** Editing yields after gradual shortening of SERPINA1 ON from 59 nt (v117.59) to 31 nt (v117.83, $n = 2$-8). Respective ON symmetry is depicted after the ON name, with the following pattern: (5′-terminus length – orphan base – 3′-terminus length). **D** Dependence of the preferred 2′ modification pattern on the ON length ($n = 3$). **E** Editing at the pre-mRNA level for five different RESTORE 2.0 ON targeting three different endogenous transcripts ($n = 3$). Editing levels depend on the maturity of the transcript. **F** Effect of 2′-modification on nt position $N_{-2}$ and $N_{-3}$ on editing yield for 40 nt

SERPINA1 ON ($n = 3$-7). **G** Testing previously described 2′ block design ON[18] against RESTORE 2.0 ON containing overall the respective same amount of stereo random PS linkages and 2′-modifications ($n = 3$). **H** Editing yields of a 45 nt RESTORE 2.0 ON upon uptake into primary human hepatocytes (5′-29-1-15 symmetry) either with (v117.81) or without (v117.80) GalNAc-moiety, targeting a previously described[18] 3′-UTR site in ACTB (5′-UAG). **I** As (**G**) but with 40 nt ON (5′−29-1-10 symmetry) with (v117.83) or without (v117.82) a GalNAc-moiety. **J** Editing yields after GalNAc-mediated uptake into primary human hepatocytes of a 40 nt RESTORE 2.0 ON (5′−29-1-10 symmetry) targeting the STAT1 Y701 site (5′-UAU) with (v117.71) or without (v117.42) GalNAc-moiety. Panels H-J are the results of $n = 3$ experiments. Data in **A**, **C–J** is shown as the mean ± s.d. Each datapoint represents a biological replicate. Editing yields in **A**, **C–G** were achieved upon transfection of 50 nM ON into HeLa cells or a SERPINA1 E342K-expressing HeLa cell model.

pre-mRNA species. This was exemplified on the other two targets, GAPDH and STAT1. In these cases, we were able to determine the editing yield for two different pre-mRNA species, a longer, less spliced and thus younger species, and a shorter, further spliced, thus older pre-mRNA species. Interestingly, we found that the editing yield in the younger species was clearly reduced (by approx. 80-50%) compared to the older pre-mRNA species, which was edited to a level similar to that of the mature mRNAs, respectively. Together, this indicates that ON act mainly in the nucleus with nuclear ADAR1p110 and to a lesser extent with ADAR1p150, which is further supported by the knockdown data and to the independence of editing on interferon treatment. It also makes it highly unlikely that a shift of ADAR1p110 from the nucleus to the cytoplasm, which can occur under certain cellular stress conditions[59], explains the good performance of our ONs. Proximal to or at the orphan cytidine base, various chemical modifications, in particular non-canonical nucleobases such as nebularine[29], Benner's base Z[28] or inosine[18,25,60], have been described to enhance editing efficiency. Except for the use of inosine inside the CBT (Supplementary Fig. 4J), we did not observe editing-enhancing effects for our fully modified ON, see Supplementary Fig. 4H+I. Still, a logical consequence was to test if – beside internucleoside linkage modifications – also the 2′-modifications close to the orphan base ($N_0$) could influence editing efficiency. Structural data[25,26] shows dense interactions of ADAR with nucleotides at positions $N_{-2}$ and $N_{-3}$. We tested eleven combinations of four different 2′-modifications, e.g., 2′-OMe (mN), 2′-F (fN), DNA (dN) and 2′-MOE (oN), based on the 40 nt fully modified SERPINA1 ON (v117.82) (Fig. 4F). As expected, the nature of the 2′-modification at these two positions strongly impacted editing yield, with the modification at $N_{-2}$ dominating the effect at position $N_{-3}$. The most preferred modification pattern was 5′-CBT-mN-fN while 2′-MOE was particularly disfavored at both sites. This data provides evidence that the region 3′ to the CBT is a hotspot to improve editing efficiency.

The previously published, stereo-pure AIMer design applies two large blocks of uniform 2′-modification, one with around fifteen 2′-F nucleotides in the 5′-half of the ON and another one with fifteen 2′-OMe nucleotides in the 3′-half of the ON, interrupted only by the CBT (containing three 2′-H nt)[18]. Of note, AIMer designs require stereo-pure PS internucleoside linkages plus the incorporation of a small number (e.g., 3-4) of stereo-pure PN linkages to achieve good editing performance[18]. From the lessons we learned in this study, the use of large blocks with uniform 2′-modification, e.g., >6 nt 2′-F or 2′-O-methyl is suboptimal. Thus, we compared the "block design" reported for AIMers[18] side-by-side with our mixed 2′-modification pattern ("mixed design") in the context of our optimized 40 nt, 33 nt, and 32 nt ON targeting SERPINA1. All tested mixed and block designs were with stereo-random PS linkage modifications (Fig. 4G). Notably, the content of each 2′-ribose modification was (near) identical for each respective ON length (Supplementary Fig. 4M). At all three ON lengths, the mixed pattern clearly outcompeted the block-design, achieving 2- to 3-fold higher editing yields. This suggests that substrate recognition by endogenous ADAR is disturbed by large blocks of uniform 2′-

modification. Among others, 2′-F ribose does not act well as an H-bond acceptor[61], and the majority of interactions of ADAR´s dsRBD with the substrate duplex are H-bonds[27]. Also, large blocks of 2′-OMe modifications can strongly reduce protein binding[62], which could also decrease substrate binding by ADAR. Deeper testing showed that interrupting the large 2′-F block had less influence on editing performance than interrupting the 2′-OMe modification block around the CBT (Supplementary Fig. 4K–L). The comparison further suggests that a proper mix of 2′-modifications can at least partially substitute the use of stereo-pure linkage chemistry.

Furthermore, we applied the established principles to test free uptake and GalNAc-assisted ON uptake into primary human hepatocytes (Fig. 4H–I). Specifically, we designed 45 nt and 40 nt long optimized ONs with or without a GalNAc-moiety targeting the same 5′-UAG codon on a previously described 3′-UTR site in the ACTB transcript[13,18]. For both ON lengths, adding the GalNAc-moiety increased the potency of the ON compared to free uptake by approximately 10-fold. Notably, compared to the previously described, stereo-pure AIMer design[18], our ON performed equally well in terms of maximum editing efficiency (>80% yield) but exhibited an approximately 6-fold higher potency ($EC_{50}$ 22–25 nM, compared to $EC_{50}[ACTB$-42$] \approx 166.2$ nM in ref. 18) indicating that potent and efficient editing ON can be designed that do not require stereo-pure PS modification. Finally, we designed a 40 nt optimized ON to edit the regulatory phosphotyrosine (Tyr701>Cys, 5′-UAU codon) in the endogenous STAT1 transcript in primary human hepatocytes by means of GalNAc-mediated delivery (Fig. 4J). Again, the GalNAc ligand clearly improved the potency of the RESTORE ON.

## In vivo editing of human SERPINA1 E342K in the PiZZ mouse model

Finally, we assessed the in vivo performance of a 40 nt fully modified ON to correct α-1-antitrypsin (A1AT) deficiency in a murine model bearing the respective human SERPINA1 PiZZ (E342K) gene. Patients suffering from the PiZZ allele exhibit excessive buildup of mutated A1AT in the liver due to protein misfolding causing either liver cirrhosis, or, due to a lack of A1AT in their blood stream and lung tissue, causing COPD[63]. For in-vivo proof-of-concept, we selected the 40 nt fully modified ON (v117.82, 5′- 29-1-10) which elicited robust editing yields of approximately 40-50% in HeLa cells in vitro. PiZZ mice were treated with a single dose of lipid nanoparticle (LNP)-encapsulated v117.82 ON via tail-vein injection in doses ranging from 0–5 mg/kg (Fig. 5A). Liver and blood serum samples were collected 3 days post treatment to evaluate editing levels and human wildtype A1AT (PiMM) serum levels, respectively. Editing yields were dose-dependent and peaked at an average of 25.8% ± 6.2% at the 5 mg/kg dose (Fig. 5B). Corrected (healthy) A1AT serum levels were determined with a wild-type- and human-specific A1AT ELISA. A1AT blood levels correlated strongly with the respective editing yields (Fig. 5C) achieving an average of 15.6 ± 7 µM at an ON dose of 3 mg/kg, which is above the serum A1AT threshold of 11 µM that is considered therapeutically meaningful in the respective protein replacement therapy[64]. Overall,

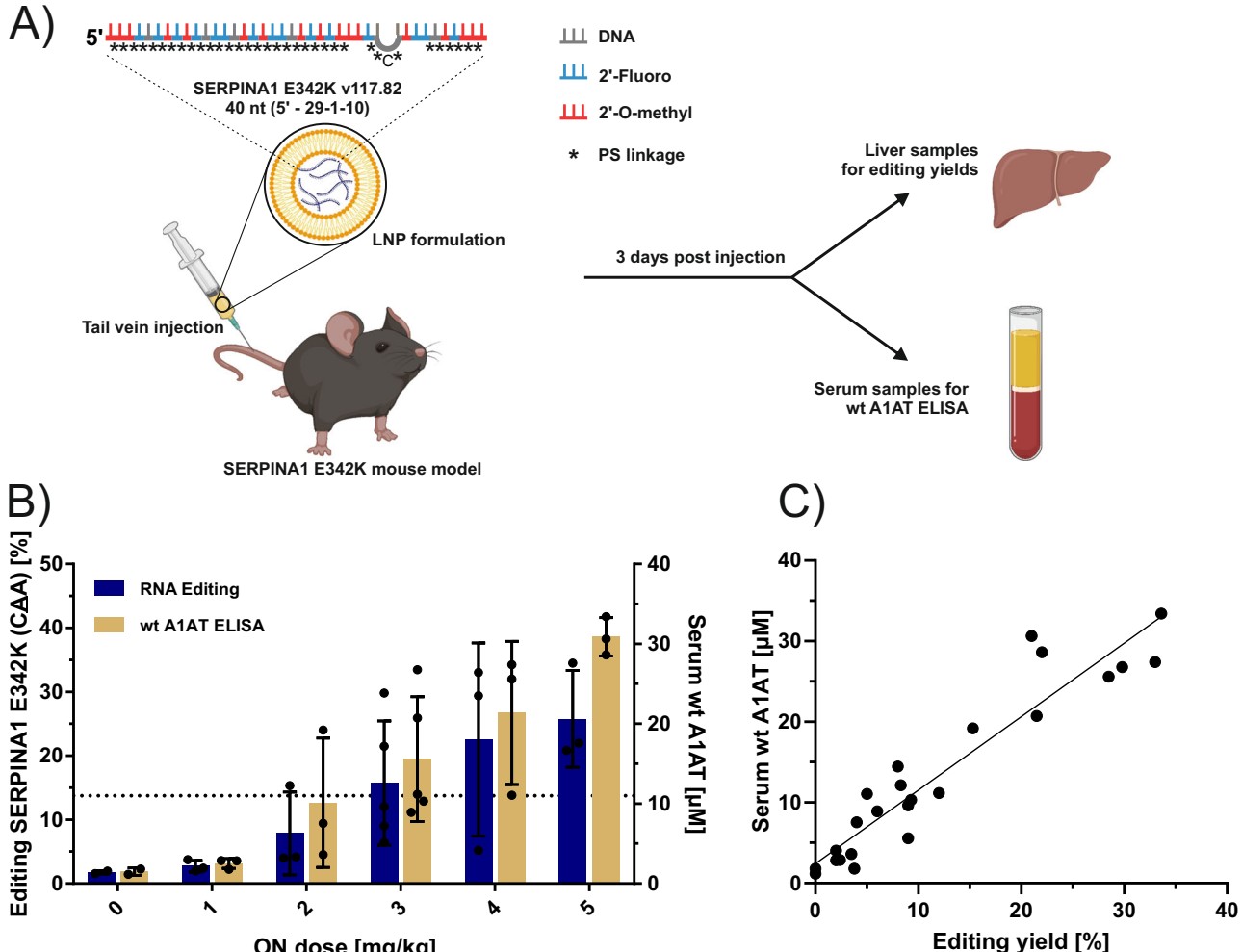

**Fig. 5 | Proof-of-concept of RESTORE 2.0 ON to reverse a common A1AT deficiency-causing SERPINA1 mutation in a murine model in vivo. A** Treatment scheme of the PiZZ (SERPINA1 E342K) A1AT deficiency mouse model. Mice were treated with a single dose of 0–5 mg/kg LNP-encapsulated SERPINA1 E342K-targeting ON v117.82 (40 nt, 5'-29-1-10 symmetry, 5'-C<u>A</u>A) via tail vein injection. LNP formulation was based on the FDA-approved siRNA drug Onpattro. Mice were sacrificed 3 days post injection. **B** Sanger editing yields from hepatocytes (blue) and serum A1AT (human, wildtype) concentrations (yellow) clearly show a dose dependent correction at RNA and protein level. The considered therapeutic threshold of 11 µM serum A1AT (ref. 64) is depicted as the dotted horizontal line. **C** Correlation of in vivo editing yields and resulting wildtype A1AT serum concentration. Data in **B** is shown as the mean ± s.d., $N$ = 2–5 independent animals per condition. Figure 5A contains an icon created in BioRender (Stafforst, T. (2025), https://BioRender.com/zfhahgx).

this demonstrates that RESTORE 2.0 ON can be applied in vivo to harness endogenous ADAR for efficient editing of a severe, disease-causing mutation in a human gene.

## Discussion

In the last decade, notable efforts have been made to develop site-directed RNA base editing for therapeutic purposes. However, many approaches still face challenges that limit their therapeutic potential. Particularly promising are RNA base editing approaches that harness endogenous ADAR and that require only the delivery of a genetically encoded guide RNA or chemically modified ON. While genetically encoded guide RNAs have lately undergone impressive optimization[14–17,44,45], major weaknesses remain regarding editing efficiency, precision, and dependency on viral delivery. Chemically modified oligonucleotides are an alternative that enable high efficiency, high precision, that avoid viral delivery, and that may allow a desirable safety profile through control of ON drug (re)dosing. Nevertheless, the design of optimal chemically modified ON for ADAR-based RNA base editing is currently underexplored and available ON designs, like the AIMers, depend on complex modifications, e.g., stereo-pure backbone modifications, that are inaccessible for broad academic and industrial

research. Here, we present the development of basic design principles that lead to the RESTORE 2.0 platform, which enable efficient and precise RNA base editing with chemically modified, unstructured ON as short as 33 nt. Importantly, our ON rely on commercially available ON modifications also found in clinically approved siRNAs and gapmers such as PS/PO, 2´-F, 2´-H, and 2´-OMe, and avoid difficult-to-synthesize stereo-pure linkage modifications.

Industry and academia has undergone enormous efforts to carve out the design rules for therapeutic siRNAs and gapmers[65]. While we can build on this knowledge, we must consider RNA base editing as its own modality and that efficient ADAR recruitment follows different rules than the recruitment of Ago2 or RNaseH. Furthermore, while the cleavage site for an siRNA or gapmer can be freely selected on the target, ADAR recruiting ON are confined to the area and sequence around the target adenosine, suggesting a greater effort is necessary to define target-agnostic rules. Still, certain sequence-independent ON design rules from this and previous studies can already be defined. First, ON with short lengths (e.g., below 40 nt) strongly prefer asymmetric designs with a short 3´-terminus[18,25,26], and ON have specific, clearly preferred lengths (e.g., 32, 33 nt) and symmetries (5´ – 26-1-6/26-1-5, Fig. 4C) for a given target. Second, fully modified ON designs

require mixtures of different 2´-ribose modifications (e.g., 2´-F, 2´-OMe, maybe including 2´-MOE and others in the future). Importantly, ON benefit from the avoidance of continuous blocks of the same 2´-modifications (e.g., more than 6 nt, Fig. 4G, Supplementary Fig. 4K - M compared to ref. 18). To some degree, the preferences of the ribose modification might depend on the ON length (Fig. 4A, D, Supplementary Fig. 4A+D+F). While there are areas in the ON where the specific 2´-modification is less important (Fig. 4A), we identified a highly modification-sensitive hotspot 3´-adjacent to the CBT at positions −2 and −3. Here, the placement of 2´-OMe and 2´-F modification at position −2 and −3 is clearly preferred (Fig. 4F, Supplementary Fig. 4K). Third, a minimum threshold of PS modifications is required for efficient endogenous ADAR recruitment (Figs. 1C, 3A, ref. 18). Again, we identified specific positions near the CBT where PS modification is less well accepted and where placement of PO, other linkage modifications or a specific linkage stereoisomer may be required (e.g., linkages g and h in Fig. 3 and ref. 18). These rules already allow the rational design of ADAR-recruiting ONs, which we call RESTORE 2.0 ON, that do not depend on contiguous stereo-pure linkage chemistry and are yet potent and metabolically stable enough for efficient editing upon relevant administration techniques such as gymnotic uptake (Fig. 2E, ref. 53), GalNAc-assisted (Fig. 4H–J, ref. 18,66) as well as LNP-mediated (Fig. 5, ref. 67) delivery, in cell lines, primary cells (Fig. 2D, Fig. 4H–J) and in vivo (Fig. 5). We also demonstrate that the LNP-delivered RESTORE 2.0 ON can harness murine ADAR in vivo to obtain a notable correction of a common, disease-relevant SERPINA1 mutation under restoration of relevant wildtype α-1-antitrypsin blood levels. Thus, this study shows that effective ADAR-recruiting ON can be designed by using readily available chemical modifications and by following the described design principles. We hope our ON design principles will simplify application and foster further research on RNA base editing with endogenous ADAR in academia and industry.

Limitations of the study. While the oligonucleotides designed here appear sufficiently safe for research use, the current data does not inform on preclinical or clinical safety. Accordingly, this work can only be understood as a preliminary step towards RNA base editing drugs. Only limited chemical modifications and combinations thereof have been explored here, and ample space remains for further optimization of two key factors in particular: ON efficacy and safety. There is a clear need to further metabolically stabilize the designs shown in this study (Supplementary Fig. 6), as efficacy and durability, but also toxicity and immunogenicity depend on metabolic stability[65,68], given that degradation kinetics and resulting byproducts may shape toxicity. Regarding safety, it may be recommendable to reduce the content of PS linkages and 2´-F nucleosides in future designs, as a certain toxicity has been reported for this combination in the past[48,58]. Additionally, RNA editing ON may be more susceptible to sequence-specific toxicity, given the confinement of the sequence to the vicinity of the target adenosine. Analyzing the protein interactome of RNA editing ON may also be helpful, since ON-protein binding is a known cause of toxicity[56,69,70]. Further, previous studies show that safety can also be influenced by dose, precise modification pattern, and ON structure[48,69,71]. Thus, multiple factors require more in-depth research in future preclinical and clinical studies.

## Methods

### Ethics statement
Mouse experiments were conducted under protocols approved by the Administrative Panel on Laboratory Animal Care (APLAC) of Stanford University.

### Antisense oligonucleotides
All ONs used in this study were purchased HPLC-purified from Euro-gentec (Belgium) or Biospring (Germany), or were generously provided by AIRNA Bio Inc., and were directly used. The sequences and

chemical modifications can be found in the supplementary ON sequence file (Supplementary Data 1).

### Analysis of in vitro RNA editing
RNA editing was analyzed as reported earlier[13]: Briefly, total RNA was extracted, treated with DNase I, reverse transcribed and cDNA was amplified by Taq PCR. The PCR product was purified on an agarose gel and sent for Sanger sequencing (Microsynth AG, Göttingen). Editing yields were calculated by dividing peak height of the guanosine by the sum of the guanosine and adenosine peak heights at the target site (cytidine and thymidine peak heights when reverse primer was used for sequencing).

### Editing procedure with ONs in ADAR-expressing Hek293 Flp-In cells
Procedure was performed as reported earlier[13]: ADAR gene expression was induced 48 h before ON transfection by seeding $2 \times 10^5$ of the respective ADAR-Flp-In 293 T-REx cells/well in 24-well plates with DMEM plus 10% FBS containing 10 ng/mL doxycycline for ADAR induction. For ON transfection, induced cells were detached and reverse transfected. For this, 5 pmol ON and 0.75 µl/well Lipofectamine 2000 (ThermoFisher Scientific) were each diluted in 10 µl OptiMEM (ThermoFisher Scientific). Both solutions were mixed and 100 µL cell suspension containing $5 \times 10^4$ cells in DMEM plus 10% FBS plus 10 ng/mL doxycycline was added onto the transfection mixture inside the 96-well plates. Cells were harvested for RNA isolation and sequencing 24 h after transfection. Results are reported in Fig. 1B.

### Editing procedure in immortalized cell lines
All cells were cultured in DMEM plus 10% FBS plus P/S, except for THP-1 which were cultured in RPMI plus 10% FBS. $1 \times 10^5$ cells/well (HeLa cells (cat. no. ATCC CCL-2), U2OS-Flp-In T-REx32 (kind donation from Elmar Schiebel), SK-NBE(2) (cat. no. ATCC CRL-2271), U87MG (cat. no. ATCC HTB-14), Huh7 (CLS GmbH, Heidelberg, cat. no. 300156), HepG2 (DSMZ, Braunschweig, Germany, cat. no. ACC180), SH-SY5Y (cat. no. ATCC CRL-2266), and A549 (European Collection of Authenticated Cell Cultures, ECACC 86012804)) were seeded in a 24-well plate. After 24 h, medium was changed (plus 3,000 U IFN-α, where mentioned) and cells were forward transfected with the ONs. Forward transfection of the ONs was achieved by diluting 25 pmol ON/well and 1.5 µl/well Lipofectamine RNAiMAX Reagent (ThermoFisher Scientific) in 50 µL OptiMEM each. Both solutions were combined after 5 min incubation and incubated for an additional 20 min before the transfection mix was distributed evenly into one well. After 24 h cells were harvested for RNA isolation and sequencing. THP-1 were transfected the same way after $3 \times 10^5$ cells/well of a 24-well plate were differentiated for 3 days in RPMI plus 10%FBS plus PMA (200 nM) and cultured for 5 days in RPMI + 10%FBS afterwards. Results are reported in Figs. 1C–E, 2A, B and Supplementary Fig. 3A–B, 4D, F, I, K, L.

### siRNA knockdown of ADAR isoforms
Procedure was performed as reported earlier[13]. Briefly, siRNA (ADAR1 (both isoforms, Dharmacon, SMARTpool: ON-TARGETplus ADAR (103) siRNA, L-008630-00-0005), ADAR1p150 (Ambion (Life Technologies), sense strand: 5´-GCCUCGCGGGCGCAAUGAAtt; antisense strand: 5´-UUCAUUGCGCCCGCGAGGCat), ADAR2 (Dharmacon, SMARTpool: ON-TARGETplus ADARB1 (104) siRNA, L-009263-01-0005) or mock (Dharmacon, siGENOME Non-Targeting siRNA Pool #2, D-001206-14-05)) was reverse transfected into HeLa cells in a 12-well format. Reverse transfection was achieved by diluting 2.5 pmol siRNA/well and 3.5 µl HiPerFect (Qiagen) in 100 µL OptiMEM each. Both solutions were combined after 5 min incubation and incubated for an additional 20 min. The transfection mix was then distributed evenly into one well before adding $1.2 \times 10^5$ HeLa cells in 800 µl DMEM plus 10% FBS on top. Medium was changed every 24 h. For ON transfection, cells were

detached 48 h after siRNA transfection. ONs were reverse transfected by mixing 0.5 µl Lipofectamine 2000 (ThermoFisher Scientific) and 5 pmol ON/well in 10 µl OptiMEM each. Both solutions were combined after 5 min incubation, incubated for an additional 20 min then distributed evenly into one well. $5 \times 10^4$ of the detached cells in 100 µL DMEM plus 10% FBS were added on top of the transfection mix. Cells were harvested for RNA isolation and sequencing 24 h after ON transfection. Results are reported in Fig. 1F.

### Editing procedure in human primary cells

All primary cells were purchased from Lonza. Normal human astrocytes (NHA, Lonza cat. no. CC-2565) were cultured in ABM Basal Medium (Lonza cat. no. CC-3187) with AGM SingleQuot Kit Supplementary & Growth Factors (Lonza cat. no. CC-4123), human retinal pigment epithelial cells (H-RPE, Lonza cat. no. 00194987) were cultured in RtEBM Basal Medium (Lonza cat. no. 00195406) supplemented with RtEGM Retinal Epithelial Cell Growth Medium SingleQuots Supplements and Growth Factors (Lonza ca. no. 00195407) without FBS (for seeding FBS was added and after 24 h medium was changed to FBS-free medium). The transfection procedure was performed the same way as for immortalized cell lines with $1 \times 10^5$ cells per 24-well seeded and 25 pmol ON transfected. Results are reported in Fig. 2D.

### Gymnotic uptake

$2 \times 10^3$ cells/well of HeLa cells were seeded into 96-well plates in the DMEM + 3% FBS. After 24 h, the medium was replaced with 100 µL fresh medium plus 20 µL ON in OptiMEM to achieve the indicated ON concentrations. Cells were harvested for RNA isolation after 3, 5, 7 or 10 days. Results are reported in Fig. 2E.

### SERPINA1 E342K editing

HeLa cells with genomically integrated SERPINA1 E342K cDNA were obtained as previously described[13]. Briefly, the SERPINA E342K cDNA obtained from HepG2 cells was genomically integrated into HeLa cells via the piggyBac transposase system and selected for 2 weeks in DMEM plus 10% FBS containing 10 µg/ml puromycin. $1 \times 10^5$ SERPINA1 E342K HeLa cells/well were seeded in 24-well plates. Cells were forward transfected with the ONs 24 h after seeding by diluting 25 pmol ON/well and 1.5 µl/well Lipofectamine RNAiMAX Reagent (Thermo Fisher Scientific) in 50 µl Opti-MEM each followed by 5 min incubation at RT. The mixtures were combined and incubated for an additional 20 min, then distributed into the respective well. Cells were harvested for RNA isolation and sequencing 48 h post transfection. Results are reported in Figs. 2F, 3A, 4A, C–F and Supplementary Fig. 4A, H, J, K.

### A1AT ELISA of SERPINA1-HeLa cell supernatant

Supernatants were collected from SERPINA1 E342K editing samples just before lysis for RNA isolation, centrifuged and stored at −20 °C. The α-1-antitrypsin (A1AT) content of each biological replicate was determined in technical triplicates with an A1AT-ELISA protocol adapted from the Teckman Lab[72] (Saint Louis University). Briefly, multi-well plates (Greiner Microlon® High Binding) were coated with polyclonal rabbit anti-human A1AT (A0012, Agilent Technologies), then blocked with 1 x PBS + 5% non-fat milk. A serial dilution of purified human A1AT and the collected cell culture supernatants were diluted in Dilution Buffer (1 x PBS + 0.1% Tween20 + 2% BSA) and added to the multi-well after the blocking step. Goat anti-human A1AT-HRP conjugated antibody (Bethyl Laboratories, A80-122P) was used as a secondary antibody (1:10,000). Wash steps in between the antibody sandwich construction were performed with 1 x PBS + 0.05% Tween20. OPD substrate (ThermoFisher, 34006) was used for detection according to manufacturer's protocol. The reaction was stopped with $H_2SO_4$ before measuring at 490 nm in a TECAN Spark® 10 M plate reader. Absolute absorption values were normalized to the values of

wildtype SERPINA1 + non-targeting ON (100%) and E342K mutant SERPINA1 + non-targeting ON (0%). The non-targeting ON was the GAPDH ORF v117.19 ON. Results are reported in Fig. 2F.

### Stability assay of ONs

15 pmol of the respective ON were diluted in 100% FBS or lysosomal solution, as noted. The lysosomal solution consisted of mixed gender rat liver tritosomes (tebu-bio, article no. 098R0610.LT) diluted to an acid phosphatase concentration of 0.1115 U/ml with a 20 mM sodium citrate solution (pH 5.0). Mock samples contained 15 pmol ON diluted in PBS only. All samples were incubated at 37 °C for the given time points, then frozen in $fN_2$ and stored immediately at −80 °C. Denaturation of the samples was achieved by adding 5 µl RNA loading dye (1:10 dilution of Rotiphorese® Sequencing gel buffer concentrate in Rotiphorese® Sequencing gel diluent, Carl Roth) each and incubation at 70 °C for 2 min. For samples containing 100% FBS, a proteinase K digestion was performed prior to the addition of RNA loading dye by adding 30 mM Tris-HCl (pH 7.5) and 60 µg proteinase K (20 U/mg, Analytik Jena) to a final sample volume of 15 µl. The digestion mix was incubated for 5 min at 50 °C. Denatured samples were then loaded on a urea (7 M) polyacrylamide (15%) electrophoresis (PAGE) gel and run for 4–6 h at 1200 V in TBE buffer. Bands were visualized through a SYBR™ Gold Nucleic Acid Gel Stain (Thermo Fisher Scientific) according to manufacturer's instructions and scanned at the excitation wavelength $\lambda_{ex} = 473$ nm with a Fujifilm FLA-5100 Fluorescent Image Analyzer. Results are reported in Figs. 2C, 3B, 4B and Supplementary Fig. 3A–B, 2B,C,E,G.

### Carrier-free ON uptake in primary human hepatocytes

Primary human hepatocytes (PHH) were purchased from Lonza (Lot no. HUM182041) and thawed, seeded, and cultured according to the manufacturer's protocol. For the free uptake of the ON, 100 µl fresh hepatocyte plating medium containing the ON at the indicated concentrations was added to 50,000 cells per well 1 h post seeding. PHHs were lysed 24 h after seeding, and further processed to analyze the RNA editing yields as described above. Results are reported in Fig. 4H–J.

### LNP formulation for SERPINA1 E342K in vivo editing

LNPs were formulated using a NanoAssemblr Benchtop microfluidic device (Precision NanoSystems Inc.). The aqueous phase consisted of ON v117.82 dissolved in 25 mM sodium acetate (pH 4). The ethanol phase consisted of DLin-MC3-DMA (MedKoo, BioFine), cholesterol (Sigma-Aldrich), 1,2-distearoyl-sn-glycero-3-phosphocholine (DSPC; MedKoo), and 1,2-dimyristoyl-rac-glycero-3-methoxypolyethylene glycol-2000 (DMG-PEG 2000; Avanti Polar Lipids) in ethanol at a molar ratio of 50:38.5:10:1.5. The N/P ratio (i.e., molar ratio of MC3 to phosphate and phosphorothioate units in the ON) was 3.45:1. The aqueous and ethanol phases were mixed by NanoAssemblr at a flow rate ratio of 3:1 and a total flow rate of 12 mL/min. The LNPs were buffer exchanged into Dulbecco's phosphate-buffered saline (DPBS; Gibco) and concentrated using 10 kDa MWCO Amicon Ultra centrifugal filters (Millipore) at 4 °C, 3200–3500 × $g$. Concentrated LNP stocks were filter-sterilized using a 0.2 µm syringe filter and stored at 4 °C. LNPs were analyzed by dynamic light scattering using a Zetasizer system (Malvern Panalytical). The particle sizes for two independent LNP preps were 79.8 nm and 80.3 nm, and polydispersity index values were 0.114 and 0.096, respectively. The encapsulation efficiency was measured using the RiboGreen method[73] with a RiboGreen Assay Kit (Life Technologies); the encapsulation efficiencies were 96% and 95% for the two LNP preparations.

### Animal experiments

Mice were housed on a 12:12 light-dark cycle, with ad libitum access to food and water at 21 °C/52% humidity. Mice expressing the human

*SERPINA1*[E342K] transgene in C57BL/6 J background were described previously[74] and were provided by Prof. Jeffrey Teckman's laboratory (Saint Louis University). Mice homozygous for the human transgene (PiZZ) were used in all experiments. Eight-to-ten-week-old male and female mice were treated with 1–5 mg/kg ASO-LNPs or DPBS (0 mg/kg) via tail-vein injection. After 3 days, blood samples were collected from the submandibular vein. The mice were subsequently euthanized by cervical dislocation under isoflurane anesthesia, and liver tissue was harvested. Liver fragments were stored in RNAprotect Tissue Reagent (Qiagen) at 4 °C. Mouse serum was collected by centrifugation of blood samples at 4 °C, 2000 × *g* for 10 min and stored at −20 °C.

### In vivo RNA editing
Mouse liver fragments were homogenized using a BeadBug 3 homogenizer (Benchmark Scientific), and total RNA was extracted using an RNeasy Mini kit (Qiagen). After DNA removal by Turbo DNase (Invitrogen), RNA was reverse transcribed using the iScript cDNA Synthesis kit (Bio-Rad). PCR was carried out using Phusion Flash High-Fidelity PCR Master Mix (Thermo Scientific). The PCR product was purified using NucleoSpin Gel and PCR Clean-Up kit (Macherey-Nagel), and analyzed by Sanger sequencing (Azenta, Elim Biopharmaceuticals).

### A1AT ELISA of blood serum
ELISA was performed according to established protocols from Prof. Mark Brantly's (University of Florida) and Prof. Jeffrey Teckman's (Saint Louis University) laboratories with minor modifications. A Nunc Max-iSorp flat-bottom 96-well plate (Thermo Fisher) was coated with 100 μL mouse anti-human wildtype-A1AT monoclonal antibody[75] (Brantly lab, University of Florida, PMID: 23195820) at a concentration of 0.167 μg/mL in Vollers buffer (15 mM sodium carbonate, 35 mM sodium bicarbonate, 3 mM sodium azide; pH 9.7). After 24–48 h incubation at 4 °C, the plate was washed with PBS buffer (pH 7.4, Gibco) containing 0.1% (v/v) TWEEN 20 (Sigma-Aldrich) ('TPBS'). Dilutions of serum samples and human A1AT standard (Athens Research & Technology) were prepared in buffer consisting of PBS (pH 7.4, Gibco), 0.1% (v/v) TWEEN 20, and 5 mg/mL BSA (Fisher) and were applied in triplicate. Standard dilutions ranged from 0.625 ng/mL to 10 ng/mL, and sample dilutions ranged from 50,000–400,000-fold. The plate was incubated for 2 h at 37 °C and washed with TPBS. Goat anti-human A1AT antibody-HRP conjugate (Bethyl Laboratories, A80-122P) was applied at 0.5 μg/mL in PBS (100 μL) and incubated at room temperature for 1 h with gentle shaking (50 rpm). After washing with TBPS, signal was developed by loading 100 μL of Sigma*Fast*™ OPD substrate (Sigma-Aldrich), dissolved in 20 mL water. After a 4 min incubation in the dark, the reaction was stopped with 50 μL of 3 M sulfuric acid. The signal was measured at 490 nm by Varioskan LUX microplate reader (Thermo Fisher) and analyzed using GraphPad Prism 10 with asymmetric sigmoidal interpolation.

### Data analysis
The analysis of RNA editing yields from sequencing traces was performed as previously described[13,17]. Briefly, adenosine-to-inosine editing yields were quantified by retrieving the signal intensity of the guanosine and adenosine peaks at the respective site and dividing the guanosine peak height by the sum of the guanosine and adenosine peak heights. Peak heights were retrieved from SnapGene (version 4.2.11). If the reverse primer was used for sequencing, cytidine and thymidine peaks were treated accordingly. Other data was analyzed using Excel 2016 and GraphPad Prism 8 if not mentioned otherwise. Figures were created with CorelDraw 2017. The manuscript was written with Word 2016.

### Statistics and reproducibility
Experiments determining editing yields via Sanger sequencing were mostly done in *n* = 2–3 biological replicates, in rare cases in *n* < 3 (max. *n* = 22). Data points are displayed individually in the column or bar graphs wherever possible. For the in vitro A1AT ELISA: each biological replicate was measured in technical triplicates. For the animal experiment data: the evaluation of the editing yields and wildtype A1AT serum levels was done with *n* = 2–5 mice per group. Each editing yield data point was measured individually as one technical replicate containing samples from multiple different liver lobes. To obtain the correlation of in vivo editing yield and A1AT serum level, samples from the same animal were used. For the SILAC-MS/MS data: experiments were performed in duplicates with swapped SILAC labels and checked correlation. Blinding was performed during the downstream analysis (sample preparation and measurement). For the RNA seq data: analysis was performed with two biological replicates per sample; the required sequencing depth was determined in a previous study[8]. For all experiments, if not specifically mentioned otherwise: No statistical method was used to predetermine sample size, group sizes for cell culture and animal experiments were selected based on the prior knowledge of variation. No data was excluded from the analyses. All experiments could be reproduced, as shown in the manuscript, and both number and nature of replicates are provided in the respective figure captions. Besides the SILAC-MS/MS experiment, the investigators were not blinded to allocation during experiments and outcome assessment due to the involvement of several experimenters. All samples were treated according to the same protocols side-by-side with the respective controls, therefore the experiments were not randomized.

### Reporting summary
Further information on research design is available in the Nature Portfolio Reporting Summary linked to this article.

## Data availability
This manuscript provides Supplementary Information on further controls (Supplementary Figs. 1– 8), tables depicting the PCR primer combinations, (Supplementary Tables 1–7) and maps of the used plasmids (Supplementary Figs. 9–10) and reference sequences (Supplementary Sequences 1–6). Source data on the RNA editing levels of each graph and uncropped stability assay gel images are provided in the Source Data file of this paper. The mass spectrometry data generated in this study have been deposited in the ProteomeXchange Consortium via the PRIDE partner repository under accession code PXD066763. The RNA seq data generated in this study has been deposited in the GEO database under accession code GSE303437. Source data are provided with this paper.

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

## Acknowledgements

We thank the University of Tübingen, the Deutsche Forschungsgemeinschaft (STA 1053/3-1, STA 1053/3-2, STA 1053/7-1, STA 1053/10-1, STA 1053/11-1, STA 1053/14-1 to T.S.), the European Research Council (CoG no. 647328, PoC no. 101069246 to T.S.), German National Academy of Sciences Leopoldina (grant no. LPDS 2019-06 to P.V.), the Deutsche Forschungsgemeinschaft (DFG, German Research Foundation) under Germany's excellence strategy (EXC 2180, 390900677: image-guided and functionally instructed tumor therapies (T.S.), Stanford University Department of Genetics, Stanford SPARK Program and Maternal & Child Health Institute for continuous financial support of our RNA editing research (to P.V., I.J. and J.B.L.). We thank the NIH for generous support to J.B.L. (NIGMS R35GM144100) and M.A.K. (NCI CA277059). We thank the NIH for generous support to M.A.K. (NIH NIAID 4R37AI071068). M.H.B. received funding from the Stanford School of Medicine Dean's Fellowship and the Stanford Propel Scholar program. We thank AIRNA Corporation for generous support with the ACTB ONs. We thank Mirita Franz-Wachtel and Boris Macek from the Proteome Center Tübingen for their support with mass spectrometry measurements and analysis of the dataset. We thank Julia Z. Adamska, all members of the Stafforst and the Li Lab for support in- and outside of the experiments shown here. Figures 1A and 5A contain icons from biorender.com (Stafforst, T. (2025), https://BioRender.com/zfhahgx. We acknowledge support from the Open Access Publication Fund of the University of Tübingen.

## Author contributions

L.S.P., T.M., and T.S. conceived and analyzed the overall study. L.S.P., T.M., N.L., L.G., C.S., S.G., D.T.H., D.F., C.L. and V.D. performed and analyzed the in vitro experiments. I.J., J.M.G., M.H-B., F.Z., P.V., M.A.K. and J.B.L. planned, performed and/or analyzed in vivo experiments. L.S.P. and T.S. wrote the manuscript. All authors proofread the manuscript.

## Funding

## Competing interests

L.S.P., T.M., I.J., N.L., P.V., J.B.L. and T.S. are inventors of patents related to RNA base editing (including WO2020001793A1, WO2022253810A1, WO2023099494, WO2024115661). I.J. and P.V. are employees of AIRNA Corporation. T.M. and C.S. are employees of AIRNA Bio Germany GmbH, a wholly owned subsidiary of AIRNA Corporation. P.V., T.M. J.B.L. and T.S. are cofounders and shareholders of AIRNA Corporation. J.B.L and T.S. are consultants to AIRNA Corporation. The remaining authors declare no competing interests.
