## [Transparent Peer Review file · Nature Communications]

Stereo-random Oligonucleotides Enable Efficient Recruitment of ADAR in vitro and in vivo

Corresponding Author: Professor Thorsten Stafforst

Version 0:

Reviewer comments:

Reviewer #1

(Remarks to the Author)

The authors detail a significant advance in RNA editing using a systematic approach to determine the optimal ON length, modification of ON chemistry to increase durability to escape lysosomal degradation and demonstrate these improvements in an actual disease model.

In future work it would be nice to document the impact of SERPINA1 E342K edit on the reduction of liver polymer of the "Z" murine model.

Minor Critique: line 373 allele is misspelled.

Reviewer #2

(Remarks to the Author)

Pfeiffer and Merkle et.al. Comments to the author

Prior work from the Stafforst laboratory developed a programmable oligonucleotide platform to recruit endogenous ADAR enzyme for efficient RNA base editing. In continuing this work, Pfeiffer and Merkle et al. adds incrementally to the understanding of the SAR of the design of guide strands for RNA editing and for alpha 1 antitrypsin show that editing about 25% of the cellular RNA may be sufficient to be of therapeutic value. The RNA editing experiments appear to be well conducted, thorough and reproducible, providing compelling evidence of the potential of RESTORE 2.0 as a versatile platform for RNA editing. However, there are significant deficiencies in the manuscript that make it unsuitable for Nature Communications:

1. Inadequate studies on ON stability

The stability of both unmodified and modified oligoribo- and oligodeoxynucleotides has been thoroughly studied. It is known that these molecules are degraded by endo- and exonucleases present in extracellular fluids, at the cell surface and throughout the cell. Yet rather than using a standard cell homogenate assay, the authors use lysosomal extracts followed by denaturing gel electrophoresis of very poor quality. I expect to see a full gel with size standard and degradates clearly separated from the intact ON with quantitation of all the degradates and the intact ON. Based on a great deal of past work, one can guess what the degradation pattern might look like, but one should not have to guess. These patterns of degradation are of significant interest because they provide information about the intramolecular structures of the ONs, the nucleases involved and metabolites, some of which may have unexpected activities and potential toxicities.

2. Limited in vivo experiments

The authors showed that they could get good editing in cells with gal-nac conjugated ONs, but used LNP formulated ONs with tail vein injections. Both the LNPs and the hydrostatic pressure of tail vein injections have proven to be problematic and largely abandoned in the ASO and siRNA space. The doses are fairly high and the variability observed high making the dose response curve difficult to trust fully. I would much rather see a gal-nac ON injected SQ using a repeat dosing schedule. This would be vastly more therapeutically relevant and generate more compelling data.

3. Lack of mechanistic insights.

The authors incorporate thoroughly studied ON modifications and the two most important modifications, PS and 2'F have well studied pleiotropic effects. PS modifications enhance nuclease resistance and, more importantly, dramatically alter interactions with proteins, enhancing the affinity and promiscuity of protein binding. Protein binding of PS containing ONs has been shown to vary as a function of the number and placement of PS moieties and the relative hydrophobicity of the 2' modifications used with more hydrophobic 2' modifications like LNA or cEt binding more proteins and with higher affinity. As the most hydrophobic 2' modification, 2'F greatly enhances protein binding. Protein binding, in turn drives ON behavior including absorption, distribution, cell uptake, subcellular distribution, cytotoxicity, adaptive and innate immune activation and pharmacodynamic effects.

The effects of the modifications studied on interactions with ADAR, stability, cellular uptake and distribution would address the question of the MECHANISMS by which the preferred designs enhance editing. For example, how much of the improvement is due to enhance stability, greater cell uptake, appropriate subcellular distribution or enhanced interactions with the target RNA or ADAR. Without these mechanistic insights, the field is actually minimally advanced in accord with traditional SAR screening approaches that represent random walks through chemical space.

ON structures are another area of interest because the intramolecular structures of ONs affect many of their properties. Are some of effects of asymmetric designs due to changes on intramolecular structures and influence on stability, cell uptake, RNA hybridization or ADAR? Simple Tms would add a lot of value and begin to address basic questions. I am also concerned that some of the sequences in some of the designs may form intermolecular structures including matrices. All these structures are influenced by modifications with higher affinity 2' modifications tending to form more and intramolecular structures as well as inter-molecular structures. Once again, this area has been well studied and there are multiple methods available to study these possibilities. Once again, I would argue that this is an important issue for the field to understand or it will take advantage of the medicinal chemistry done for ASOs, but repeat many of the mistakes that delayed ASO development.

4. Lack of Off-target analysis

In previous work, Merkle et al., demonstrated ADAR1p150-sensitive RESTORE 1.0 ONs to have very few off-target editing and minimal changes in editing homeostasis of HeLa cells treated with and without IFN- α . However, the new platform appears to preferentially recruit ADAR1p110 isoform in both the long and short design. ADARp110 differs from ADAR1p150 by its constitutive expression and nuclear localization and therefore may introduce global cell homeostatic changes when recruited by exogenous RESTORE 2.0 ON. To demonstrate the safety profile and therapeutic application of this system when delivered to cells, the authors need to include genome-wide off-target analysis of the RESTORE 2.0 platform.

5. Lack of ADAR protein analysis

In Fig 1E: While it may not be surprising that editing efficiencies of GAPDH ORF vary between cell lines, these observations could be strengthened by a western blot of ADAR1 (both isoforms) and ADAR2 protein expression across the lines. Variations in ADAR levels could perhaps be correlated with editing efficacy and help explain the observed differences between short vs long design editing across the cell types. This correlation may also inform the therapeutic approach of the platform based on target cell/tissues and ADAR expression levels.

6. Failure to focus on therapeutic index

The author's stated purpose and the reason the manuscript might be of interest to Nature Communications is to report advances that enhance the therapeutic utility of guide strands for RNA editing. Therapeutic advances are made when the therapeutic index is enhanced. Yet, the authors never asked about potential off target effects, cytotoxicity, innate immune activation or provide any evidence that their preferred designs have an appropriate therapeutic index. PS, 2'F modified ONs have been shown to be very toxic in cells, animals and man. The mechanisms proven include, rapid loss of paraspeckle proteins and apoptosis, nucleolar toxicity and degradation to the 2'F nucleosides that can be incorporated into DNA and RNA via the nucleoside salvage pathway. Patisiran, a PS and 2'F modified siRNA was shown to cause hepatotoxicity and metabolic acidosis in patients, a pattern eerily similar to that of FIAU toxicities. Though only limited sites were studied with LNA modifications, LNA ASOs have proven to be very hepatotoxic, renal toxic and immunostimulatory in man at therapeutic doses.

If the authors want to claim that their designs are potentially therapeutically valuable, they need to prove that, particularly since they chose to use 2'F as a key modification.

Minor comments:

1. "side" in the abstract should be "site" I think.
2. Fig 1F: The editing efficiency of GAPDH ORF by ON v120.2 is estimated to be ~40% in no siRNA or mock siRNA conditions in HeLa cells. However, unlike ON v117.19 this is significantly less than that observed in Fig1E where ~80% editing of GAPDH ORF was observed by the same ON. Authors should clarify differences between the two experiments, if any, to describe the variance in editing efficiencies between experiments. As presented in Fig 1F, the observed editing efficiency is not as significant compared to RESTORE 1.0 (Fig 1E).
3. Fig 2A-B: Authors should specify in main text or figure legend which cell line was used for this assay.
4. Fig 2F: ON v117.19 (top) is depicted with five 2'-O-methyl modified bases immediately adjacent to the 5' of the CBT. I believe these were meant to be unmodified, as described in Supplementary Table 1.
5. On page 9, the authors state "Nearly every tested pattern...(v117.59 to v117.77 in Fig. 4A) was stable in lysosomes (Supplemental Fig. 2C)." However, lysosome stability data was not provided for all candidates in Suppl. Fig 2C nor does it

- appear in Suppl. Fig 2B (only includes v117.39-117.58). Given the authors are evaluating new modifications on stability, lysosome data should be included as similarly shown in Fig 4B or appended in the Supplementary as stated.
6. Related to the above, the authors have made major improvements on SERPINA1 E342K-targeting ON editing efficiency and stability in FBS and lysosome from Fig 2F to Fig 4F. The impact of this editing change on A1AT protein levels should be evaluated as previously done (Fig 2F) for lead candidates, as increased A1AT levels would strengthen the impact of the new designs.
 7. Fig 4F: v117.167-v117.170 have 1 extra nucleotide following the CBT. As drawn, the sequence symmetry is 25-1-7, not 25-1-6 as described in Supplementary Table 2.
 8. Supplementary Fig 2C: SERPINA1 ON id: ISIS116847 is undefined and not mentioned in text. What does this data point represent?
 9. Supplementary Fig 2C appears pixelated when zoomed in, making it difficult to read the text.

Reviewer #3

(Remarks to the Author)

Review of „Improved Oligonucleotide Design Enables Efficient Recruitment of ADAR in vitro and in vivo.“ Laura S. Pfeiffer

Site-directed RNA base editing (SDRE) by ADAR is being developed to correct human disease mutations at the level of the mutant primary transcript or mRNA. Many therapies will have to be targeted to post-replicative cell types, whereas CAS9-mediated mutation correction requires the DNA repair activity in replicating cells. ADAR-mediated RNA editing can also potentially avoid risks from off-target genome mutagenesis, although a disadvantage is that SDRE will probably have to be applied to patients repeatedly. Also, efficiently and specifically redirecting endogenous ADAR enzymes to target mutations in endogenous transcripts is not easy.

The authors report on how they have developed RESTORE 2.0 ON oligonucleotide-directed ADAR1 RNA editing a) using significantly shorter oligonucleotides with b) improved efficacy, and c) stability against nuclease digestion. They go on to show correction of pathogenic point mutations efficiently after GalNAc-mediated uptake into human primary hepatocytes, and proof of in-vivo efficacy in mice, using only commercially available, stereorandom chemical modifications. These authors previous approach, termed RESTORE (1.0), used 95 nt long RNA oligonucleotides (ON) comprised of a 40 nt long specificity domain (mediating programmable binding to the target mRNA) plus a 55 nt dsRNA hairpin-forming ADAR-recruiting domain. In RESTORE 2.0 they have removed the ADAR-recruiting dsRNA hairpin to shorten the RNA oligonucleotides for easier synthesis and tried to improve the efficiency of the target RNA-binding to compensate for loss of ADAR recruitment through the dsRNA hairpin.

The SDRE method here aims to harness mainly ADAR1, which is much more widely expressed across the body. Normally, it is mainly ADAR2 that is associated with highly efficient site-directed RNA editing at specific bases in endogenous transcripts, and structural information available is for ADAR2. ADAR1 normally binds to and edits longer dsRNAs promiscuously at efficiencies below 10% at any particular base, although it does edit a few sites with higher efficiencies. A 59 nt unstructured ON targeting a 5'-UAG stop codon in the ORF of the human GAPDH transcript in HEK293 gave efficient editing in cells stably overexpressing different ADAR isoforms, when compared to the original RESTORE 1.0 ON 124 (v25, Fig. 1B). Based on symmetric (5'-29-1-29) and asymmetric (5'-48-1-10) positionings of the orphan cytosine over the target adenosine they also tested the recruitment of endogenous ADAR in HeLa cells (instead of overexpressed ADAR isoforms in the HEK293 Flp-In cells). Editing by recruiting endogenous ADAR1 was possible only when applying a high phosphotiorate (PS) content to the ONs during synthesis, to increase the intracellular stability of the ON. However, placing PS too close to the central base triplet where the ON has an orphan cytidine base over the target adenosine prevents RNA editing.

The unmodified ONs had low stability in FBS and, since RNase A, the major RNase in FBS, has a clear preference to attack RNA at pyrimidine sites they resorted to adding modifications at pyrimidines in the ONs, particularly 2'-F on the ribose, at pyrimidines only. These were more stable in FBS and gave efficient editing after gymnotic transfer into different primary cells, RPE, NHA and NHBE. Blocks of the same 2'-OH modification or repetitive patterns like alternate base modification harmed editing more than mixed, random patterns.

They then transferred the partial stabilization strategy to a human disease-relevant setting; repair of the SERPINA1 E342K mutation (5'-CAA codon context), a common cause of α -1-antitrypsin (A1AT) deficiency, in a HeLa cell model overexpressing the mutated cDNA. Besides Sanger sequencing to measure editing, they analyzed the total A1AT protein levels from the cell supernatant using an antibody ELISA. The partially 2'-F modified design (v117.25, 46.7% editing) clearly outperformed the unstable parent ON (v117.19, 15.3% editing, Fig. 2F). They investigated which internucleoside linkage (a to j) around the CBT area (nucleotide positions N+4 to N-5) is amenable for PS modification (see Fig. 3A). This is unclear because in Fig. 3 the a to j superscripts are too small and bases on both sides of the orphan C appear to be labelled -1 to -5. PS linkages 5' of the CBT do not affect editing; these 5' ON bases would be under the ADAR2 dsRBD on the 3' side of the edited A. PS insertion during synthesis at bases 3' of the CBT do reduce editing; these bases would be under the catalytic deaminase domain (DD) on the 5' side of the edited A where many DD-base protein contacts occur.

Stability of ONs in a commercially available lysosomal solution purified from rat liver tritosomes was tested and found to differ substantially from FBS. Pyrimidine-modified ONs were now not stable, perhaps because lysosomal nucleases like RNase T2 prefer to cleave after purines at purine-pyrimidine interfaces. So they went back and showed that modifying both purine and pyrimidine ribose 2' hydroxyls randomly, apart from bases where ADAR2 contacts ribose 2' hydroxyls, with mixtures of-H or F modifications, stabilized the ONs in lysosomal solution and improved editing. They also managed to get editing with 32 or 33 nucleotide ONs corresponding to the minimal footprint of ADAR2 dsRBD2+DD on dsRNA. They applied the newly established principles to test free uptake and GalNAc-assisted uptake of RESTORE 2.0 ON into primary human hepatocytes; GalNAc substantially improved the potency of the ONs.

Finally, they assessed the capacity of a 40 nt RESTORE 2.0 ON fully modified ON (v117.82, 5'- 29-1-10) which elicited

robust editing yields of approximately 40-50% in HeLa cells in vitro to correct α -1-antitrypsin (A1AT) deficiency in mice bearing the human SERPINA1 PiZZ (E342K) gene PiZZ mice were treated with a single dose of lipid nanoparticle (LNP)-encapsulated v117.82 ON via tail-vein injection in doses ranging from 0 - 5 mg/kg (Fig. 5A). Patients suffering from the PiZZ mutation exhibit excessive buildup of mutated A1AT in the liver due to protein misfolding causing either liver cirrhosis, or, due to a lack of A1AT in their blood stream and lung tissue, causing COPD. PiZZ mouse liver and blood serum samples were collected 3 days post treatment to evaluate editing levels and human wildtype A1AT (PiMM) serum levels, respectively. A1AT blood levels correlated strongly with the respective editing yields (Fig. 5C) achieving an average of $15.6 \pm 7 \mu\text{M}$ already with an ON dose of 3 mg/kg, which is above the threshold of 11 μM serum A1AT that is considered a clinically meaningful therapy response.

It would be interesting to know whether the ONs are acting in the nucleus or in the cytoplasm. ADAR1 has a constitutive nuclear p110 isoform and a cytoplasmic p150 isoform that is interferon-induced. The authors are trying to get ADAR1 p110 SDRE without inducing interferon or expression of the ADAR1 p150 isoform. It is not clear that inducing ADAR1 p150 and getting higher SDRE in transfected cells or mice would do any harm. Treating HeLa cells or with interferon after treatment with 40 nt fully modified ON (v117.82, 5'-29-1-10) or adding polyI:C should increase the SDRE efficiency. Interferon treatment should improve the SDRE in mice also.

Also, Roy Parker's group have shown that treatment of cells with poly I:C causes ADAR1 p110 to move out of the nucleus. Therapeutic ONs will form dsRNA with target mRNA and it would be interesting to see whether these dsRNAs induce some highly-expressed ISG transcripts or might also draw some ADAR1 p110 out of the nucleus. Maybe this is required for ADAR1 p110 SDRE? Separating nuclear and cytoplasmic fractions would show if there is an increase in cytoplasmic ADAR1 p110 in response to RESTORE 1 or RESTORE2 ONs with or without modified bases.

RESTORE 2.0 ON may simplify the industrial and academic translation of RNA base editing into clinical applications as the principles elucidated should apply to retargeting sites in general. However, one suspects that stereopure oligonucleotides will eventually be preferred if synthesis difficulties are overcome commercially

Reviewer #4

(Remarks to the Author)

In this manuscript, Pfeiffer, Merkle, and Colleagues aim to enhance and establish guidelines for the design of guides for the site-directed RNA editing system RESTORE. Compared to RESTORE 1.0, the Authors significantly reduce the length of the guide RNA constructs and test various combinations of chemical modifications to enhance stability and editing rates. In particular, they focus on optimizing the system through modifications that are already clinically approved and cost-effective to synthesize. Additionally, the Authors demonstrate GalNAc mediated uptake of their modified ONs and show its effectiveness in vivo via injection of lipid nanoparticles.

Overall, the work presented here is well performed and an interesting read. It will surely be useful to the entire community interested in site-directed RNA editing.

Major Comments

- rather than an update of the RESTORE system, this seems to me a complete overhaul of the system. What made the RESTORE system unique was the double stranded handle to facilitate ADAR recruitment. In its current form, there is little that makes the system conceptually different from other ADAR-recruiting systems, be they developed in an academic or in a company environment.

- while the amount of combinations tested is massive and the Authors try to show the linear progression of the work, the development of the 4 guides (GAPDH, ACTB, SERPINA1, STAT1) could have proceeded independently from each other, as not all 'things' apply/are tested to all the guides. While the take-home message is "mix and match the different modifications", with some rationale behind it, not all rules can be generalized: e.g., the shortening of the guide significantly affects SERPINA1 editing (Fig 4C). If the aim of the work was to find the rules to design optimal guides, it might have been better to test all guides under the same conditions (taking into account forced differences - e.g. different base composition of the targets).

- While the results shown in the animal are quite promising, it would make sense to include in the experiments other ONs to understand the benefits of the modification patterns in the animals. Would it be possible to evaluate the degradation of the ONs in the animals?

Minor Comments

- It seems that the major deaminase involved is the p110 ADAR1 isoform, which is mainly nuclear. Are the ONs imported into the nucleus? Could this be one factor to be taken into account?

- I understand the choice of ON names, due to the large number of ONs tested. Yet, they are hard to follow. For example, the v117.19 name applies to two different targets within the same figure. Finding a more indicative name could help the reader keep better track of the tested ONs (e.g. target - # of 5' bases - # of 3' bases . version).

- Figures are clear and easy to read. Having color-coded ON illustrations makes it easier to understand the modifications. However, some panels could be arranged to make better use of space. This could even provide more space to enlarge some graphs.

- Sometimes it seems as there is a whole lot of experiments that did not make the cut into the manuscript, yet their outcome has had an effect on the flow of the manuscript. For example, the sentence at lines 140-141 tells of modified bases at the CBT, but these experiments are never shown.

- Figure 1E and from line 147 - It is mentioned that RESTORE 1.0 achieves 25% editing only with IFN- α . However, RESTORE 1.0 is only shown tested on HeLa. The results for the other cell lines are not present in the supplementary figures.

- What is the source of the variability in editing efficiency among the different cell lines? Could the Authors speculate on this?

- Line 351 – The paragraph feels like it was cut short. The wording makes it look like there was more regarding the GalNAc mediated ON.
- line 41 - it is not clear whether the “ref” in parentheses points to the cited references (3, 4) or it is just a missing reference .
- line 71 - the BBZ acronym does never appear beyond this point
- line 137 - the PS acronym is explained after the first appearance of it (line 136)

Reviewer #5

(Remarks to the Author)

Version 1:

Reviewer comments:

Reviewer #1

(Remarks to the Author)

I believe the authors have thoroughly addressed the concerns of this and other reviewers within the scope of this manuscript.

Reviewer #2

(Remarks to the Author)

I thank the authors for their responses , some of which were helpful, particularly the added data on stability. However none of the comments or arguments made address my concern. As i indicated my concern is that the data do not support the claims made by the authors in the title and throughout the manuscript.

Having been involved in RNA targeted therapeutics for >35 years and have published hundreds of papers in this space, I think I have gained insight into the incremental nature of progress in establishing a new drug discovery platform from a blank piece of paper and all the modifications the author's use were first studied in the antisense space long before they were used in siRNAs. I certainly have the scars to prove that I understand and am sympathetic to the incrementalism required. I am equally well aware of harm done by exaggerated claims based on inadequate data. Even some of the exaggerated claims made in the early days of RNA targeted drug discovery can be forgiven because nothing was known. That is not the case today. There is a great deal known about the challenges and the risks posed by modifications the modifications chosen by the authors.

I will say this as bluntly as I can say it : I will not accept a manuscript making the claims made without data addressing the obvious issues I raised. I am deeply concerned that even more naive efforts will use KNOWN TOXIC nucleotide modifications in guide strands based on this paper and set the field back at least a decade.

2. I appreciate the nuclease digestion data, but do find them a bit hard to accept fully based on the known properties of 2'F nucleosides. More important , to CLAIM that the new designs are better in vivo , one must show safety and PK properties. Based on the data available, I could accept that the designs enhance the performance in vitro and may , if safer modifications are found, be a step toward better in vivo performance, but author steadfastly refuse to narrow their claims.

3. LNPs have been largely abandoned in the ASO/Si RNA space for very good reasons. If the authors want to claim in vivo improvement, they need to show editing with the gal-nac. Gal-nac conjugation delivers ONs via a different mechanism than LNPs and gal-nac conjugation results in lower ON concentrations in hepatocytes than LNP. I could get into many other differences , but to use LNPs when gal-nacs are available is a giant leap backwards.

4. The mechanistic insights I would like to see are those that would DIRECT future SAR work. I am pleased that authors are evaluating protein interactions, but they offer no insights of value. (By the way, it has been clearly shown that the gapmer structure itself is not a major factor in protein binding or the formation of aggregates. Rather the lipophilicity of 2' modifications and the PS number and placement are the major factors).

5. With high affinity modifications like 2'F and LNA, traditional folding programs are useless, but the authors could easily map the structure and should because the structure may provide insights into mechanism and challenges ahead.

6. Finally, irrespective of the type of institution in which work is performed , I have to believe that the authors would agree that the conclusions and claims made in a manuscript must be fully supported by the data.

Reviewer #4

(Remarks to the Author)

We thank the authors for the detailed and well-articulated responses to the reviewers' comments. We believe the changes improve an already good work.

I would also apologise for the poor wording of the first comment: the comment did not imply that the work was not interesting, it only meant to highlight the fact that what had made RESTORE stand out among similar linear ONs, like AIMers, was the inclusion of the ADAR recruitment domain and that removal of the domain strips away this distinguishing feature. As such, it

might make sense to come up with a new name rather than call it RESTORE 2.0. It also does not sound fair to the other approaches the attempt to name a somehow identical strategy (despite all development behind it) as an update of a strategy that had fairly distinctive features.

Minor comments:

- Reference 46 (Yi et al. Genome Biology (2023) 24:243) could be included in the introduction. Lines 51-52
- Line 119 typo – written “from” instead of “form”
- Fig 3 – Text mentions N0 for orphan cytosine but figure shows C0.
- Line 363 pre-mRNA and mature mRNA comparison – maybe include a bit more detail about the target transcripts. Seems too vague the way it is.
- Line 486 should be “build” instead of “built”
- Line 506 “were” can be removed

Reviewer #5

(Remarks to the Author)

Version 2:

Reviewer comments:

Reviewer #6

(Remarks to the Author)

The article by Pfeiffer et al. reports on the chemical optimization of short oligonucleotides (30–40 nt), demonstrating their efficiency in inducing robust editing both in vitro and in vivo (for a single target).

Developing fully chemically modified ADAR-recruiting oligonucleotides remains an area of significant research and commercial interest. Although such oligonucleotides have been published recently, achieving maximum potency typically requires the use of stereoselective PS and PN modifications, which are not yet widely available commercially.

The authors show that altering the 2'-O-methyl and 2'-fluoro modification pattern from a block design to a mixed design (i.e., a mixmer of 2'-O-methyl and 2'-fluoro residues) enhances editing efficiency in vitro. However, data is presented for only one concentration.

While the overall data quality is acceptable, the manuscript would need substantial rewriting to be suitable for a general audience.

Major Comments

Outdated Context and Lack of Positioning Against Prior Work

The article appears to compile data accumulated over several years, presented from a historical perspective with additional content likely added in response to reviewer feedback. However, much of the experimental content is outdated given recent developments in the field. In particular, the work should be positioned clearly in the context of Monian et al., published in Nature Biotechnology (March 2022) [<https://www.nature.com/articles/s41587-022-01225-1>], which reported fully chemically modified short ADAR-targeting oligonucleotides.

Key Innovation Is Not Clearly Highlighted

Within this current landscape, the most significant contribution of the manuscript is not the development of fully chemically modified ADAR-targeting oligonucleotides (which has been done previously), but rather the demonstration that highly active ADAR-recruiting ASOs can be engineered without the need for stereoselective modifications. This makes the technology more broadly accessible to the research community. This is a valuable and important insight, but the message is currently diluted by the inclusion of extensive historical data and less relevant details.

Use of Confusing Terminology (“Restore-1”, “Restore-2”)

The use of marketing-like terminology such as “Restore-1” and “Restore-2” is confusing and detracts from scientific clarity. While the intention to brand or classify the compounds is understandable, it adds little scientific value and may confuse general readers. It would be preferable to use clear, descriptive terms such as “fully chemically modified ADAR-recruiting oligonucleotides.” If the authors wish to introduce new terminology (e.g., “Restore”), it should be clearly defined in the discussion section as a proposed nomenclature for future reference. It's worth noting that in the siRNA field, compounds are typically referred to by their modification profiles, unless part of a marketed or proprietary pipeline.

Lack of Critical Evaluation of Study Limitations

The manuscript lacks a balanced discussion of the study's limitations. For example, the claim that “Restore-2 is safe” is not adequately supported by the presented data. At best, the available data shows that in a single study, LNP-mediated delivery of the ADAR-recruiting ASO led to efficient editing three days post-treatment, without observable toxicity or animal mortality.

Regarding the safety of 2'-fluoro modifications: both the reviewers' and authors' points have merit, but the truth lies in the nuance. Toxicity of 2'-F-modified oligonucleotides in vivo depends on dose, modification pattern, and overall structure. In particular, concerns have been raised about protein binding and mitochondrial toxicity from high doses (~25 g/year, revusiran failed PHASE III) of fully modified siRNAs lacking 5'-terminal backbone stabilization. If the authors wish to support their claims, this would require in vivo evaluation of degradation kinetics using LC-MS (and I can confidently predict significantly lower stability compared to fully modified siRNAs), as well as analysis of clearance metabolites and exaggerated pharmacology—areas that fall outside the scope of this report.

While the oligonucleotides designed here appear sufficiently safe for research use—which is an important contribution—the current data does not inform on preclinical or clinical safety. Moreover, as single-stranded oligonucleotides, these ADAR-recruiting ASOs are (1) subject to known sequence-specific ASO toxicities, and (2) significantly less stable than duplex siRNAs, potentially resulting in different toxicity profiles upon repeated dosing.

None of this detracts from the scientific importance of the findings—but it does warrant a more objective presentation of the data and its implications.

Version 3:

Reviewer comments:

Reviewer #7

(Remarks to the Author)

The authors have sufficiently addressed the comments of Reviewer 6. These include references and comparisons to current state of the art AIMer technology, clear focus on the key innovation of the study, and a thorough evaluation of the limitations of the study. As such, I believe this manuscript is appropriate for publication.

One note. In lines 520-521 the authors say that analyzing the protein interactome will be helpful to understand the protein interactome. Please rephrase.

Rebuttal Letter

REVIEWER COMMENTS

Reviewer #1 (Remarks to the Author):

The authors detail a significant advance in RNA editing using a systematic approach to determine the optimal ON length, modification of ON chemistry to increase durability to escape lysosomal degradation and demonstrate these improvements in an actual disease model.

In future work it would be nice to document the impact of SERPINA1 E342K edit on the reduction of liver polymer of the "Z" murine model.

Minor Critique: line 373 allele is misspelled.

RESPONSE: We thank the reviewer for this very positive feedback on the manuscript. The typo was corrected and the hint well taken.

Reviewer #2 (Remarks to the Author):

Pfeiffer and Merkle et.al. Comments to the author

Prior work from the Stafforst laboratory developed a programmable oligonucleotide platform to recruit endogenous ADAR enzyme for efficient RNA base editing. In continuing this work, Pfeiffer and Merkle et al. adds incrementally to the understanding of the SAR of the design of guide strands for RNA editing and for alpha 1 antitrypsin show that editing about 25% of the cellular RNA may be sufficient to be of therapeutic value. The RNA editing experiments appear to be well conducted, thorough and reproducible, providing compelling evidence of the potential of RESTORE 2.0 as a versatile platform for RNA editing. However, there are significant deficiencies in the manuscript that make it unsuitable for Nature Communications:

RESPONSE: We thank the author for acknowledging the quality of our data and the compelling evidence. With the new data added, we now touch all aspects of the criticism brought up by the reviewers. However, we also wish to mention that this screening effort only provides a starting point to develop ADAR harnessing drugs. None of the molecules included here are mature enough to become drug candidates without further optimization and characterization. More importantly, we provide the industry and academia with a well-defined starting point to develop such drugs from the commercially accessible and clinically approved, stereorandom RNA modification space. Furthermore, we also define some general and translatable rules, some also based on structural insights, and give some further mechanistic insight how RESTORE ON function. For example, we now provide clear evidence that editing with RESTORE 2.0 ONs happens co-transcriptionally, in parallel to splicing in the nucleus. We show that RESTORE 2.0 ONs are located in the nucleus and interact with typical nuclear RNA binding proteins. We show that the modification of the oligonucleotide helps to abolish ISG induction, likely due to an evasion of sensing of the RESTORE ON by the relevant innate immune sensors. While writing this, the first clinical data proved that harnessing of endogenous ADAR with oligonucleotides has been provided (<https://www.science.org/content/article/buoyed-milestone-clinical-result-rna-editing-poised-treat-diseases>). This modality will now make its way into the clinic and it is important that our data is now and visibly published to foster this development even if not all questions are answered to full clarity today.

1. Inadequate studies on ON stability

The stability of both unmodified and modified oligoribo- and oligodeoxynucleotides has been thoroughly studied. It is known that these molecules are degraded by endo- and exonucleases present in extracellular fluids, at the cell surface and throughout the cell. Yet rather than using a standard cell homogenate assay, the authors use lysosomal extracts followed by denaturing gel electrophoresis of very poor quality. I expect to see a full gel with size standard and degradates clearly separated from the intact ON with quantitation of all the degradates and the intact ON. Based on a great deal of past work, one can guess what the degradation pattern might look like, but one should not have to guess. These patterns of degradation are of significant interest because they provide information about the intramolecular structures of the ONs, the nucleases involved and metabolites, some of which may have unexpected activities and potential toxicities.

RESPONSE: We have now added the primary data (full urea PAGE gels) of examples from all stages of RESTORE 2.0 development, starting from the unmodified 59 mer, to the partially modified 59 mer, to the fully modified 59 mer and finally the fully modified 40 mer, Supplementary Fig. 1 and 6. The gels show that single nucleotide resolution of the degradation products is achieved. As FBS and lysosome extract have different nucleases, mainly DNaseI and RNaseA (cutting after pyrimidines, preferably uridine) and RNaseT1/2 (cutting at purine/pyrimidine interfaces, preferably between GU), respectively, the degradation patterns look very different. We are aware that a deeper understanding of the weakest bonds will help to further improve the pharmacological properties of the ON, for example may increase the duration of action, limit immunogenicity, etc. We added some discussion about this to the manuscript. However, it is not the scope of the manuscript to fine-tune a drug candidate, optimize its stability, toxicity or immunogenicity, but rather to break the path towards such a development. Thus, in the manuscript we stay at the stage of defining the further engineering challenge.

2. Limited in vivo experiments

The authors showed that they could get good editing in cells with gal-nac conjugated ONs, but used LNP formulated ONs with tail vein injections. Both the LNPs and the hydrostatic pressure of tail vein injections have proven to be problematic and largely abandoned in the ASO and siRNA space. The doses are fairly high and the variability observed high making the dose response curve difficult to trust fully. I would much rather see a gal-nac ON injected SQ using a repeat dosing schedule. This would be vastly more therapeutically relevant and generate more compelling data.

RESPONSE: We fully agree, the targeting of the liver with subcutaneously administered GalNAc ON is the ultimate goal. However, within the scope of this manuscript we are not able to perform further experiments in this animal model. However, we wish to mention that this data on LNP-formulated ADAR recruiting ON is the first peer-reviewed proof-of-concept in this model of human disease. Even though working on it, competing companies are currently not sharing any such data in sufficient detail to replicate their experiments. Thus, we believe that the current data is already very meaningful for the field in academia as well as industry.

3. Lack of mechanistic insights.

The authors incorporate thoroughly studied ON modifications and the two most important modifications, PS and 2'F have well studied pleiotropic effects. PS modifications enhance nuclease resistance and, more importantly, dramatically alter interactions with proteins, enhancing the affinity and promiscuity of protein binding. Protein binding of PS containing ONs has been shown to vary as a function of the number and placement of PS moieties and the relative hydrophobicity of the 2' modifications used with more hydrophobic 2' modifications like LNA or cEt binding more proteins

and with higher affinity. As the most hydrophobic 2' modification, 2'F greatly enhances protein binding. Protein binding, in turn drives ON behavior including absorption, distribution, cell uptake, subcellular distribution, cytotoxicity, adaptive and innate immune activation and pharmacodynamic effects.

RESPONSE: We are very aware of these aspects. Indeed, we have recently developed an assay to detect the intracellular protein interactome at pharmacologically relevant gapmer concentrations (<https://www.nature.com/articles/s41589-023-01530-z>). We have started applying this approach also to RESTORE 2.0 ON and added some new data that compares the interactome of two 40 mers with varying degree of PS over PO, 49% and 87%, see Supplementary Fig. 7, respectively. The interactome of both binders includes many candidates known from the gapmer space, including NONO, SFPQ, HSP90. The interactome (e.g. the proteins that have been enriched) of the RESTORE ON was independent of the PS content, however, the enrichment factors were slightly higher for the PS-rich ON, which is in accordance with a tighter binding. During this assay, we also looked at the intracellular ON distribution in a concentration-dependent manner. Gapmers tend to form larger structures in the nuclei, including paraspeckle-like structures. For the RESTORE ON we see a clear nuclear localization with a rather uniform distribution in the nucleoplasm, somewhat avoiding the nucleoli (while some toxic LNA-flanked gapmers tend to accumulate in the nucleoli). At high concentrations nuclear foci start to appear, which may be a sort of PS-bodies and/or paraspeckle-like structures. 2'-F gapmers have been described to be toxic by degradation of NONO. We do not find any hints thereof in the context of two tested RESTORE 2.0 ONs, Supplementary Figure 7D. Clearly, this is only the beginning of a deeper mechanistic understanding, which we, however, believe is out of the scope of the manuscript. We wish to mention though that Waves's AIMer platform uses a stretch of 15 (!) 2'-F/PS nucleotides and it was successfully applied in NHPs without any sign of toxicity at levels where editing was well observed.

The effects of the modifications studied on interactions with ADAR, stability, cellular uptake and distribution would address the question of the MECHANISMS by which the preferred designs enhance editing. For example, how much of the improvement is due to enhance stability, greater cell uptake, appropriate subcellular distribution or enhanced interactions with the target RNA or ADAR. Without these mechanistic insights, the field is actually minimally advanced in accord with traditional SAR screening approaches that represent random walks through chemical space.

RESPONSE: We fully agree with the reviewer's comment. However, to get a full picture and understanding requires time. It took >20 years for the siRNA and gapmer field to arrive there. We are looking into all these aspects. However, the scope of this manuscript is an initial screening in order to find a robust starting point for the design of efficient ADAR recruiting oligonucleotides that are accessible to everyone and that use only clinically approved chemical modifications. We will cover many of the indicated aspects in future work. To give you and the reader some first insights, we have now added some data regarding the intracellular ON distribution (it is localized and active in the nucleus), ON stability, the intracellular ON protein binding and that the ON do not induce severe NONO degradation as observed for 2'-F modified gapmer ASOs, see Supplementary Fig. 6 and 7. However, a really meaningful treating of the different pharmacological aspects of the ON is out of the scope of the current manuscript, which is already very rich in data and content.

ON structures are another area of interest because the intramolecular structures of ONs affect many of their properties. Are some of effects of asymmetric designs due to changes on intramolecular structures and influence on stability, cell uptake, RNA hybridization or ADAR? Simple Tms would add a lot of value and begin to address basic questions. I am also concerned that some of the sequences

in some of the designs may form intermolecular structures including matrices. All these structures are influenced by modifications with higher affinity 2' modifications tending to form more and intramolecular structures as well as inter-molecular structures. Once again, this area has been well studied and there are multiple methods available to study these possibilities. Once again, I would argue that this is an important issue for the field to understand or it will take advantage of the medicinal chemistry done for ASOs, but repeat many of the mistakes that delayed ASO development.

RESPONSE: The reviewer is absolutely right, secondary structure formation of the ON / guide RNA or of the target RNA can negatively affect target engagement and editing efficiency. In particular for the long, encodable guide RNAs like LEAPER and CLUSTER guide RNAs, this is an important topic and we now regularly apply in silico structure optimization during the design process. The shorter an ON gets, the less likely is the chance of inhibitory secondary structure formation. We have had several examples where small changes in the guide RNA symmetry had large effects on editing efficiency. To rule out that the induction of inhibitory secondary structure is causing these effects, we now provide some folding analysis of poor versus well performing RESTORE 2.0 ONs and discuss this in the main text. Briefly, we do not see in such examples that their secondary structures can explain the differences in their editing performance, see also Supplementary Fig. 5.

4. Lack of Off-target analysis

In previous work, Merkle et al., demonstrated ADAR1p150-sensitive RESTORE 1.0 ONs to have very few off-target editing and minimal changes in editing homeostasis of HeLa cells treated with and without IFN- α . However, the new platform appears to preferentially recruit ADAR1p110 isoform in both the long and short design. ADARp110 differs from ADAR1p150 by its constitutive expression and nuclear localization and therefore may introduce global cell homeostatic changes when recruited by exogenous RESTORE 2.0 ON. To demonstrate the safety profile and therapeutic application of this system when delivered to cells, the authors need to include genome-wide off-target analysis of the RESTORE 2.0 platform.

RESPONSE: We have now added some insightful off-target data by comparing the unmodified with the partially modified RESTORE 2.0 ON in primary NHA cells, see Supporting Fig. 2. The unmodified ON induced a clear ISG response and a measurable change in the natural editing homeostasis towards higher global editing levels (190 of 14500 detected natural editing sites were significantly differently edited, almost all upregulated). In contrast, the partially modified RESTORE 2.0 ON did not induce an ISG response and did not give rise to a change in the natural editing homeostasis. This is well in agreement with literature knowledge, which says that chemical modification of pyrimidine nucleotides, in particular uridine nucleotides, helps to mitigate ISG activation. We added a respective discussion.

5. Lack of ADAR protein analysis

In Fig 1E: While it may not be surprising that editing efficiencies of GAPDH ORF vary between cell lines, these observations could be strengthened by a western blot of ADAR1 (both isoforms) and ADAR2 protein expression across the lines. Variations in ADAR levels could perhaps be correlated with editing efficacy and help explain the observed differences between short vs long design editing across the cell types. This correlation may also inform the therapeutic approach of the platform based on target cell/tissues and ADAR expression levels.

RESPONSE: The editing levels fluctuate quite a lot in cell culture. We speculate that the main effect is the transfection yield rather than the ADAR amount. We tried in the past to determine ADAR1 protein levels by Western blot between cell lines but it turned out that it is technically challenging to compare among cell lines. For example, we were not able to normalize in a meaningful manner with cell numbers, or GAPDH, or ACTB protein levels. It is also questionable how meaningful this is for the

later translation. In cell lines, ADAR2 is not expressed while this is the case in many tissues in vivo. Interestingly, the editing levels in primary cells are frequently higher than in many standard cell lines. This was also the case in this study. Clearly, in the liver, the ADAR amounts are not limiting. The same seems true in RPE, NHA, and NHBE. In our recent study with CLUSTER guide RNA (Reautschnig et al., Nature Biotech 2024, <https://www.nature.com/articles/s41587-024-02313-0>), we analyzed the main factors of editing efficiency in six different brain regions and found that ADAR levels were not limiting in vivo. The main effector was the guide RNA abundance. Additionally, a previous study of the tissue-wide transcriptome from 2019 has shown there is only a poor correlation between ADAR expression and A-to-I editing levels (see Fig. 3 in <https://doi.org/10.1038/s41592-019-0610-9>). We added some discussion about this point in response to the reviewers' criticism.

6. Failure to focus on therapeutic index

The author's stated purpose and the reason the manuscript might be of interest to Nature Communications is to report advances that enhance the therapeutic utility of guide strands for RNA editing. Therapeutic advances are made when the therapeutic index is enhanced. Yet, the authors never asked about potential off target effects, cytotoxicity, innate immune activation or provide any evidence that their preferred designs have an appropriate therapeutic index. PS, 2'F modified ONs have been shown to be very toxic in cells, animals and man. The mechanisms proven include, rapid loss of paraspeckle proteins and apoptosis, nucleolar toxicity and degradation to the 2'F nucleosides that can be incorporated into DNA and RNA via the nucleoside salvage pathway. Patisiran, a PS and 2'F modified siRNA was shown to cause hepatotoxicity and metabolic acidosis in patients, a pattern eerily similar to that of FIAU toxicities. Though only limited sites were studied with LNA modifications, LNA ASOs have proven to be very hepatotoxic, renal toxic and immunostimulatory in man at therapeutic doses. If the authors want to claim that their designs are potentially therapeutically valuable, they need to prove that, particularly since they chose to use 2'F as a key modification.

RESPONSE: This is an academic paper, reporting an initial screen to define a starting point for drug development. It is certainly not meant as a full proof of drug candidate. We are aware of the potential toxicity and immunogenicity aspects of our RESTORE 2.0 ON. However, we wish to mention that the competing AIMER platform from WAVE therapeutics applies ON that contain a 15 nt (!!) uninterrupted block of 2'-F/PS, and report tolerance in NHP (<https://doi.org/10.1038/s41587-022-01225-1>). It might well be that the combination of very hydrophobic PS/DNA with the very hydrophobic PS/2'-F in gapmers is required to induce NONO degradation, nuclear ASO/protein aggregate formation and the described toxicity. As indicated above, we did not see similar aggregate formation nor did we detect notable NONO degradation with RESTORE ON. We now discuss in the manuscript that the potential toxicity of 2'-F/PS has to be taken seriously and that a reduction of either 2'-F and/or PS in future designs might be advantageous. A lower PS content leads to reduced protein binding as observed in the interactome for two RESTORE ON with differences in the PS content.

Minor comments:

1. "side" in the abstract should be "site" I think.

RESPONSE: Corrected.

2. Fig 1F: The editing efficiency of GAPDH ORF by ON v120.2 is estimated to be ~40% in no siRNA or mock siRNA conditions in HeLa cells. However, unlike ON v117.19 this is significantly less than that observed in Fig1E where ~80% editing of GAPDH ORF was observed by the same ON. Authors should clarify differences between the two experiments, if any, to describe the variance in editing efficiencies between experiments. As presented in Fig 1F, the observed editing efficiency is not as significant compared to RESTORE 1.0 (Fig 1E).

RESPONSE: We were also wondering about this result. It happens from time to time that editing yields differ among experiments even in the same cell line with the same ON. This is why we always include control editing reactions with very well established “lead ON” when we test new designs. In case of the siRNA knockdown experiment, overall editing levels were particularly low. This experiment has been very reproducible, and we have not seen this in any other experiment. We believe it might be due to the fact that a double transfection was used in this experiment (RESTORE ON transfection after siRNA transfection).

3. Fig 2A-B: Authors should specify in main text or figure legend which cell line was used for this assay.

RESPONSE: The information (HeLa cells) was now added to the caption. Thank you for the helpful hint.

4. Fig 2F: ON v117.19 (top) is depicted with five 2'-O-methyl modified bases immediately adjacent to the 5' of the CBT. I believe these were meant to be unmodified, as described in Supplementary Table 1.

RESPONSE: We thank the reviewer for very thorough reviewing. The mistake lies in the Supplementary Table and was corrected now. The 2'-O-methyl block was introduced here to block two bystander editing events.

5. On page 9, the authors state “Nearly every tested pattern...(v117.59 to v117.77 in Fig. 4A) was stable in lysosomes (Supplemental Fig. 2C).” However, lysosome stability data was not provided for all candidates in Suppl. Fig 2C nor does it appear in Suppl. Fig 2B (only includes v117.39-117.58). Given the authors are evaluating new modifications on stability, lysosome data should be included as similarly shown in Fig 4B or appended in the Supplementary as stated..

RESPONSE: We have now included some updated stability data on RESTORE 2.0 ON in response to major point 1 (see Supplementary Figures 1 and 6). We think that exemplifying the general trends on the selected ONs, which have been more broadly characterized, fully suits the purpose. It is well known from other antisense drugs like siRNAs and gapmers that a high degree of modification helps to stabilize the molecule against nuclease degradation. It is not the purpose here to provide a full picture for each individual modification pattern.

6. Related to the above, the authors have made major improvements on SERPINA1 E342K-targeting ON editing efficiency and stability in FBS and lysosome from Fig 2F to Fig 4F. The impact of this editing change on A1AT protein levels should be evaluated as previously done (Fig 2F) for lead candidates, as increased A1AT levels would strengthen the impact of the new designs.

RESPONSE: We do not believe that this data adds much at this intermediate stage during oligonucleotide optimization, also because these assays are expensive as the access to the PiZ and

PiM specific antibodies is very limited. Most importantly, we have this data, with the expensive PiM-specific antibody for the in vivo PoC where it is most relevant for this study.

7. Fig 4F: v117.167-v117.170 have 1 extra nucleotide following the CBT. As drawn, the sequence symmetry is 25-1-7, not 25-1-6 as described in Supplementary Table 2.

RESPONSE: We thank the reviewer for this very careful reading. Indeed, there was a mistake in the Figure, the data in the Supplementary Table was correct (25-1-6), the Figure was corrected now.

8. Supplementary Fig 2C: SERPINA1 ON id: ISIS116847 is undefined and not mentioned in text. What does this data point represent?

RESPONSE: ISIS116847 is a very well-studied RNaseH gapmer with 2'-MOE flanks from Ionis Pharmaceuticals against the PTEN mRNA (<https://doi.org/10.1093/nar/gku484>). It was used here as a control of an ASO drug with well-characterized and sufficient in vivo stability and activity. This was now better explained in the Figure caption, and the full sequence and modification and a reference are now given in the manuscript.

9. Supplementary Fig 2C appears pixelated when zoomed in, making it difficult to read the text.

RESPONSE: We know of this issue. Since the figure is quite large, Word compresses it once the file is saved, thereby pixelating the text. The respective editable files we have shared with the editor are of considerably better quality.

Reviewer #3 (Remarks to the Author):

Review of „Improved Oligonucleotide Design Enables Efficient Recruitment of ADAR in vitro and in vivo.“ Laura S. Pfeiffer

Site-directed RNA base editing (SDRE) by ADAR is being developed to correct human disease mutations at the level of the mutant primary transcript or mRNA. Many therapies will have to be targeted to post-replicative cell types, whereas CAS9-mediated mutation correction requires the DNA repair activity in replicating cells. ADAR-mediated RNA editing can also potentially avoid risks from off-target genome mutagenesis, although a disadvantage is that SDRE will probably have to be applied to patients repeatedly. Also, efficiently and specifically redirecting endogenous ADAR enzymes to target mutations in endogenous transcripts is not easy.

The authors report on how they have developed RESTORE 2.0 ON oligonucleotide-directed ADAR1 RNA editing a) using significantly shorter oligonucleotides with b) improved efficacy, and c) stability against nuclease digestion. They go on to show correction of pathogenic point mutations efficiently after GalNAc-mediated uptake into human primary hepatocytes, and proof of in-vivo efficacy in mice, using only commercially available, stereorandom chemical modifications. These authors previous approach, termed RESTORE (1.0), used 95 nt long RNA oligonucleotides (ON) comprised of a 40 nt long specificity domain (mediating programmable binding to the target mRNA) plus a 55 nt dsRNA hairpin-forming ADAR-recruiting domain. In RESTORE 2.0 they have removed the ADAR-recruiting dsRNA hairpin to shorten the RNA oligonucleotides for easier synthesis and tried to improve the efficiency of the target RNA-binding to compensate for loss of ADAR recruitment through the dsRNA hairpin.

The SDRE method here aims to harness mainly ADAR1, which is much more widely expressed across the body. Normally, it is mainly ADAR2 that is associated with highly efficient site-directed RNA

editing at specific bases in endogenous transcripts, and structural information available is for ADAR2. ADAR1 normally binds to and edits longer dsRNAs promiscuously at efficiencies below 10% at any particular base, although it does edit a few sites with higher efficiencies.

A 59 nt unstructured ON targeting a 5'-UAG stop codon in the ORF of the human GAPDH transcript in HEK293 gave efficient editing in cells stably overexpressing different ADAR isoforms, when compared to the original RESTORE 1.0 ON 124 (v25, Fig. 1B). Based on symmetric (5'-29-1-29) and asymmetric (5'-48-1-10) positionings of the orphan cytosine over the target adenosine they also tested the recruitment of endogenous ADAR in HeLa cells (instead of overexpressed ADAR isoforms in the HEK293 Flp-In cells). Editing by recruiting endogenous ADAR1 was possible only when applying a high phosphostiorate (PS) content to the ONs during synthesis, to increase the intracellular stability of the ON. However, placing PS too close to the central base triplet where the ON has an orphan cytidine base over the target adenosine prevents RNA editing.

The unmodified ONs had low stability in FBS and, since RNase A, the major RNase in FBS, has a clear preference to attack RNA at pyrimidine sites they resorted to adding modifications at pyrimidines in the ONs, particularly 2'-F on the ribose, at pyrimidines only. These were more stable in FBS and gave efficient editing after gymnotic transfer into different primary cells, RPE, NHA and NHBE. Blocks of the same 2'-OH modification or repetitive patterns like alternate base modification harmed editing more than mixed, random patterns.

They then transferred the partial stabilization strategy to a human disease-relevant setting; repair of the SERPINA1 E342K mutation (5'-CAA codon context), a common cause of α -1-antitrypsin (A1AT) deficiency, in a HeLa cell model overexpressing the mutated cDNA. Besides Sanger sequencing to measure editing, they analyzed the total A1AT protein levels from the cell supernatant using an antibody ELISA. The partially 2'-F modified design (v117.25, 46.7% editing) clearly outperformed the unstable parent ON (v117.19, 15.3% editing, Fig. 2F). They investigated which internucleoside linkage (a to j) around the CBT area (nucleotide positions N+4 to N-5) is amenable for PS modification (see Fig. 3A). This is unclear because in Fig. 3 the a to j superscripts are too small and bases on both sides of the orphan C appear to be labelled -1 to -5. PS linkages 5' of the CBT do not affect editing; these 5' ON bases would be under the ADAR2 dsRBD on the 3' side of the edited A. PS insertion during synthesis at bases 3' of the CBT do reduce editing; these bases would be under the catalytic deaminase domain (DD) on the 5' side of the edited A where many DD-base protein contacts occur. Stability of ONs in a commercially available lysosomal solution purified from rat liver tritosomes was tested and found to differ substantially from FBS. Pyrimidine-modified ONs were now not stable, perhaps because lysosomal nucleases like RNase T2 prefer to cleave after purines at purine-pyrimidine interfaces. So they went back and showed that modifying both purine and pyrimidine ribose 2' hydroxyls randomly, apart from bases where ADAR2 contacts ribose 2' hydroxyls, with mixtures of-H or F modifications, stabilized the ONs in lysosomal solution and improved editing. They also managed to get editing with 32 or 33 nucleotide ONs corresponding to the minimal footprint of ADAR2 dsRBD2+DD on dsRNA. They applied the newly established principles to test free uptake and GalNAc-assisted uptake of RESTORE 2.0 ON into primary human hepatocytes; GalNAc substantially improved the potency of the ONs.

Finally, they assessed the capacity of a 40 nt RESTORE 2.0 ON fully modified ON (v117.82, 5'-29-1-10) which elicited robust editing yields of approximately 40-50% in HeLa cells in vitro to correct α -1-antitrypsin (A1AT) deficiency in mice bearing the human SERPINA1 PiZZ (E342K) gene PiZZ mice were treated with a single dose of lipid nanoparticle (LNP)-encapsulated v117.82 ON via tail-vein injection in doses ranging from 0 - 5 mg/kg (Fig. 5A). Patients suffering from the PiZZ mutation exhibit excessive buildup of mutated A1AT in the liver due to protein misfolding causing either liver cirrhosis, or, due to a lack of A1AT in their blood stream and lung tissue, causing COPD. PiZZ mouse liver and blood serum samples were collected 3 days post treatment to evaluate editing levels and human wildtype A1AT (PiMM) serum levels, respectively. A1AT blood levels correlated strongly with the respective editing yields (Fig. 5C) achieving an average of $15.6 \pm 7 \mu\text{M}$ already with an ON dose of 3 mg/kg, which is above the threshold of $11 \mu\text{M}$ serum A1AT that is considered a clinically meaningful therapy response.

RESPONSE: We thank the reviewer for this very comprehensive evaluation of our data and experiments. We have now delivered a revised main Figure 3 with larger fonts in the superscript (linkage a-j) and subscript (nucleotide position +4 to -5) to improve readability.

It would be interesting to know whether the ONs are acting in the nucleus or in the cytoplasm. ADAR1 has a constitutive nuclear p110 isoform and a cytoplasmic p150 isoform that is interferon-induced. The authors are trying to get ADAR1 p110 SDRE without inducing interferon or expression of the ADAR1 p150 isoform. It is not clear that inducing ADAR1 p150 and getting higher SDRE in transfected cells or mice would do any harm. Treating HeLa cells or with interferon after treatment with 40 nt fully modified ON (v117.82, 5'-29-1-10) or adding polyI:C should increase the SDRE efficiency. Interferon treatment should improve the SDRE in mice also.

Also, Roy Parker's group have shown that treatment of cells with poly I:C causes ADAR1 p110 to move out of the nucleus. Therapeutic ONs will form dsRNA with target mRNA and it would be interesting to see whether these dsRNAs induce some highly-expressed ISG transcripts or might also draw some ADAR1 p110 out of the nucleus. Maybe this is required for ADAR1 p110 SDRE? Separating nuclear and cytoplasmic fractions would show if there is an increase in cytoplasmic ADAR1 p110 in response to RESTORE 1 or RESTORE2 ONs with or without modified bases.

RESPONSE: We now provide data that answers these questions. First, we add editing data from the pre-mRNAs of three different targets (ACTB, STAT1, GAPDH) and clearly show that editing takes place in the nucleus prior to splicing. Thus, editing is mainly performed by nuclear ADAR1p110, which fits well to the independence of IFN treatment and ADAR1 knockdown data. We also added now microscopy data, and ON-protein interactome data which shows that RESTORE ON are clearly present in the nucleus. Finally, we have added NGS/RNA seq data of modified versus partially modified RESTORE 2.0 ON in NHA (primary astrocytes). This data shows that an unmodified ON does induce an ISG response and increases ADAR editing activity globally. However, this was not the case for the partially modified ON. We have added the respective discussion(s) in the manuscript.

RESTORE 2.0 ON may simplify the industrial and academic translation of RNA base editing into clinical applications as the principles elucidated should apply to retargeting sites in general. However, one suspects that stereopure oligonucleotides will eventually be preferred if synthesis difficulties are overcome commercially

RESPONSE: We are not that sure about this. Stereopure RNA drugs will always remain much more expensive and cumbersome to manufacture. In a 30 mer, 2^{30} isomers are conceivable which might make it impossible to find the best isomer. Until today, stereopure gapmers and siRNAs optimized in vitro have not at all translated into any meaningful clinical benefit compared to the much more accessible stereorandom analogs. While in vitro assays might screen for efficiency and target engagement, protein binding and metabolic stability in human might become more important in the clinics. This said, we believe that it is a major selling point of this work to demonstrate that accessible stereorandom compounds with standard chemistry can act very well to harness ADAR.

Reviewer #4 (Remarks to the Author):

In this manuscript, Pfeiffer, Merkle, and Colleagues aim to enhance and establish guidelines for the design of guides for the site-directed RNA editing system RESTORE. Compared to RESTORE 1.0, the Authors significantly reduce the length of the guide RNA constructs and test various combinations of chemical modifications to enhance stability and editing rates. In particular, they focus on optimizing the system through modifications that are already clinically approved and cost-effective to synthesize. Additionally, the Authors demonstrate GalNAc mediated uptake of their modified ONs and show its effectiveness in vivo via injection of lipid nanoparticles.

Overall, the work presented here is well performed and an interesting read. It will surely be useful to the entire community interested in site-directed RNA editing.

RESPONSE: We thank the reviewer for this very positive feedback.

Major Comments

- rather than an update of the RESTORE system, this seems to me a complete overhaul of the system. What made the RESTORE system unique was the double stranded handle to facilitate ADAR recruitment. In its current form, there is little that makes the system conceptually different from other ADAR-recruiting systems, be they developed in an academic or in a company environment.

RESPONSE: Thank you for that comment. Scientifically, the usage of structured ADAR recruitment domains is very interesting and we continue to use and develop them in our CLUSTER approach where the guide RNA is expressed and a few nucleotides more or less do not make much difference to the size of the construct. However, in the world of highly modified oligonucleotide (ON) drugs it appears better to reduce the size of the ON to facilitate its manufacture and to better control its pharmacological properties, like protein binding and delivery. In accordance, most industry players seem to bet on single-stranded, short ON lacking an ADAR recruitment domain. However, this makes our manuscript not less but maybe even more interesting for a broad readership. Regarding academic players, we must say that there is very little data out there on the ON design to harness endogenous ADAR. Again, this study provides a starting point for many academic labs to harness endogenous ADAR efficiently with ON which are easy to access in terms of length and modification chemistry.

- while the amount of combinations tested is massive and the Authors try to show the linear progression of the work, the development of the 4 guides (GAPDH, ACTB, SERPINA1, STAT1) could have proceeded independently from each other, as not all 'things' apply/are tested to all the guides. While the take-home message is "mix and match the different modifications", with some rationale behind it, not all rules can be generalized: e.g., the shortening of the guide significantly affects SERPINA1 editing (Fig 4C). If the aim of the work was to find the rules to design optimal guides, it might have been better to test all guides under the same conditions (taking into account forced differences - e.g. different base composition of the targets).

RESPONSE: We fully understand the argument of the reviewer but clearly these several hundred ON have been tested with a certain history. As far as possible, we try to carve out some general design principles but also show the limitations of the rules. In the siRNA and gapmer field it took years and thousands of ON and dozens of targets to deduce some rules. ADAR harnessing guide RNAs are somewhat different from that. First, ADAR substrates are not perfect duplexes (e.g. like an siRNA) or perfect DNA/RNA hybrids (like for the gapmer), but natural ADAR substrates contain bulges and mismatches, and every codon as well as every target sequence may require slightly different

chemistry. For a knockdown the target site can be freely chosen throughout the whole transcript and thus more general rules can be deduced, e.g. for an ideal G/C content. This is different for the ADAR recruiting ON. Thus, there will always be more effort for optimization and only more data over time will help. We discuss this aspect now better in the revised manuscript.

- While the results shown in the animal are quite promising, it would make sense to include in the experiments other ONs to understand the benefits of the modification patterns in the animals. Would it be possible to evaluate the degradation of the ONs in the animals?

RESPONSE: The animal experiment here is a proof-of-concept. While we understand the wish of the reviewer and also see the benefit of such data, we think that this is out of the scope of the current manuscript. We will certainly consider the in vivo metabolic ON stability in future animal studies.

Minor Comments

- It seems that the major deaminase involved is the p110 ADAR1 isoform, which is mainly nuclear. Are the ONs imported into the nucleus? Could this be one factor to be taken into account?

RESPONSE: Yes they are. We have now added evidence from fluorescence microscopy that show accumulation of the ONs in the nucleus. Furthermore, we added substantial data that demonstrates that editing happens mainly at the pre mRNA, thus prior and in parallel to splicing and before nuclear export, see Figure 4E and Supplementary Fig. 7.

- I understand the choice of ON names, due to the large number of ONs tested. Yet, they are hard to follow. For example, the v117.19 name applies to two different targets within the same figure. Finding a more indicative name could help the reader keep better track of the tested ONs (e.g. target - # of 5' bases - # of 3' bases . version).

RESPONSE: We spent quite some time thinking about this when we prepared the manuscript initially and decided to stay with the historical ON nomenclature. This at least enabled to provide a clear and correct Table of sequences and modifications. The ONs are different in target, symmetry, sequence and chemical modifications pattern. Also the suggested way of naming would not be easy to follow nor would it be necessarily unambiguous. Also changing this at this stage would be a large source of error and may frustrate the other reviewers. That is why we would like to stay with it as it is.

- Figures are clear and easy to read. Having color-coded ON illustrations makes it easier to understand the modifications. However, some panels could be arranged to make better use of space. This could even provide more space to enlarge some graphs.

RESPONSE: We have rearranged the panels in Figure 4 to include the new panel E and are likely to do more so during at the galley proof stage once we have arrived there.

- Sometimes it seems as there is a whole lot of experiments that did not make the cut into the manuscript, yet their outcome has had an effect on the flow of the manuscript. For example, the sentence at lines 140-141 tells of modified bases at the CBT, but these experiments are never shown.

RESPONSE: We are not absolutely sure what the review refers to here, maybe the cited line numbers are wrong. However, the optimization of the chemistry in the CBT was done with our engineered SNAP-ADAR tool (Vogel et al., Nature Methods 2018). The corresponding rich data set was actually not yet published. This is why it comes out of the blue in this manuscript and also why we cannot refer to a publication on this.

- Figure 1E and from line 147 - It is mentioned that RESTORE 1.0 achieves 25% editing only with IFN- α . However, RESTORE1.0 is only shown tested on HeLa. The results for the other cell lines are not present in the supplementary figures.

RESPONSE: You are right, the RESTORE 1.0 ON was only tested in HeLa cells. We rewrote line 147ff to make this clearer. Thank you for that hint!

- What is the source of the variability in editing efficiency among the different cell lines? Could the Authors speculate on this?

RESPONSE: This is an interesting point and there is no absolute answer yet. We believe that the transfection bias is the main factor here. ADAR levels might also be critical, however, as discussed above, it is difficult to compare ADAR levels between different cell lines side-by-side. However, we have published recent in vivo PoC data in the murine brain and found that the editing yield was not limited by the ADAR levels in bulk tissue from different brain regions, but always by the guide RNA amounts (Reautschnig et al., Nature Biotech 2024, <https://www.nature.com/articles/s41587-024-02313-0>). We think that ADAR levels are higher in vivo (e.g. ADAR2 is expressed in vivo but not in cell lines) and are generally not limiting. This also fits well to the overall comparably good editing levels in primary cells (Fig. 2D: RPE, NHA, NHBE; and Fig. 4 G,H in primary hepatocytes). We have now added some discussion of these thoughts to the revision.

- Line 351 – The paragraph feels like it was cut short. The wording makes it look like there was more regarding the GalNAc mediated ON.

RESPONSE: Indeed, we had shortened the manuscript at this paragraph, we have now added a sentence to provide a smoother transition.

- line 41 - it is not clear whether the “ref” in parentheses points to the cited references (3, 4) or it is just a missing reference .

RESPONSE: Yes, it does, “ref” points to the cited references. “ref” was now removed.

- line 71 - the BBZ acronym does never appear beyond this point

RESPONSE: The reviewer is right. The abbreviation was introduced for a panel showing data with this base modification. As this panel was moved to the Supplementary the abbreviation is not used anymore and thus was deleted in the main text.

- line 137 - the PS acronym is explained after the first appearance of it (line 136)

RESPONSE: This was corrected now.

Rebuttal Letter

Reviewers' comments:

Reviewer #1 (Remarks to the Author):

I believe the authors have thoroughly addressed the concerns of this and other reviewers within the scope of this manuscript.

RESPONSE: We thank the reviewer for this very positive feedback on our comprehensive revision of the manuscript regarding the concerns brought up by all four reviewers.

Reviewer #2 (Remarks to the Author):

I thank the authors for their responses, some of which were helpful, particularly the added data on stability. However none of the comments or arguments made address my concern. As I indicated my concern is that the data do not support the claims made by the authors in the title and throughout the manuscript.

Having been involved in RNA targeted therapeutics for >35 years and have published hundreds of papers in this space, I think I have a keen insight into the incremental nature of progress in establishing a new drug discovery platform from a blank piece of paper and all the modifications the author's use were first studied in the antisense space long before they were used in siRNAs. I certainly have the scars to prove that I understand and am sympathetic to the incrementalism required. I am equally well aware of harm done by exaggerated claims based on inadequate data. Even some of the exaggerated claims made in the early days of RNA targeted drug discovery can be forgiven because nothing was known. That is not the case today. There is a great deal known about the challenges and the risks posed by modifications the modifications chosen by the authors.

I will say this as bluntly as I can say it: I will not accept a manuscript making the claims made without data addressing the obvious issues I raised. I am deeply concerned that even more naive efforts will use KNOWN TOXIC nucleotide modifications in guide strands based on this paper and set the field back at least a decade.

2. I appreciate the nuclease digestion data, but do find them a bit hard to accept fully based on the known properties of 2'-F nucleosides. More important, to CLAIM that the new designs are better in vivo, one must show safety and PK properties. Based on the data available, I could accept that the designs enhance the performance in vitro and may, if safer modifications are found, be a step toward better in vivo performance, but the author steadfastly refuses to narrow their claims.

RESPONSE: We are very sorry that the reviewer cannot appreciate our comprehensive new data. Overall, we get the impression that the main point here is that the reviewer has the strong opinion that the applied modifications, in particular 2'-F, are too toxic for their use in RNA drugs in general and that the results and implications of our work are thus overinterpreted. We would like to address this concern in several ways.

First, 2'-F modifications are in use in FDA-approved siRNA drugs. Specifically, the last five siRNAs that have been approved by the FDA since 2019 (Givosiran, Lumarsiran, Inclisiran, Vutrisiran and Nedosiran) all contain 2'-F nucleotides, three of some contain around 45% in the Ago2-loaded antisense strand. Alnylam has reported comprehensive data showing that also metabolites, e.g. 2'-Fluoro monomers do not accumulate and are neither inhibitors nor substrates of human polymerases (see reference 48, and newly added references 69 and 71). We discuss this aspect now more comprehensively in the revised manuscript lines 522ff.

Second, 2'-F modifications have been shown to be toxic in gapmers in combination with their very high hydrophobicity resulting from the high DNA and PS content. One important mechanism of toxicity is the sequestration of intracellular proteins and the degradation of paraspeckle proteins like NONO. Being single-stranded and PS-rich, RESTORE 2.0 ASOs might behave more like gapmers than siRNAs. However, different from gapmers, RESTORE ASOs contain a large fraction of 2'-O-methyl monomers. Precisely this combination, i.e., both 2'-F and 2'-O-methyl chemical modifications, is used in the aforementioned FDA-approved siRNAs, where lengthy in vivo studies have shown inconspicuous results concerning toxicity (such as in reference 48). For our case, we have addressed the binding properties of RESTORE ASOs in our revision and have checked for the stability of the paraspeckle protein NONO. We could not find the typical nuclear aggregation nor the degradation of NONO that has been described for gapmers. Finally, we wish to mention that Wave has published highly 2'-F/PS containing AIMer oligonucleotides that have been effective for harnessing endogenous ADAR in nonhuman primates without signs of toxicity.

The reviewer expresses the concern that our data, the deduced general design principles, and our in vivo PoC experiment lacking a full tox assessment, might direct the field to wrong directions.

First, we now clearly state in our revised discussion (lines 515ff) that RNA drug toxicities are important to consider and that we did not assess this in our PoC experiment. We now clearly say that further assessment and optimization is required to develop this approach into a clinically applicable and tolerable drug. We hope that this is sufficient tone down and pointing out of limitations of the study.

Second, we do not see the danger that we mislead the field. All RNA base editing drug programs we are aware of (AIRNA, Wave Life Sciences, KorroBio, ProQR) make already use of 2'-F/PS in their pipelines and clinical programs. The recent, huge success of demonstrating for the first time affective harnessing of human ADAR in an AATD patients by Wave Life Sciences (see newly added reference 72) shows that this is possible without facing dose-limiting toxicities. From what we know, WVE-006 contains ample of 2'-F/PS modifications. We are very sure that every effective ADAR recruiting RNA drug will contain at least a certain amount of 2'-Fluoro modification.

Overall, we hope we could clarify our main points. 2'-Fluoro is required, it is already in clinical testing, and we believe that in the context of RNA base editing, this will be actionable with reasonable tolerability/safety profile as RNA base editing oligos are different, e.g. less hydrophobic than gapmers. Furthermore, we believe that we have made the limitations of our study regarding tox assessment now very clear and and also highlight the potential concerns around 2'-F/PS, including a deeper discussion of relevant literature.

3. LNPs have been largely abandoned in the ASO/Si RNA space for very good reasons. If the authors want to claim in vivo improvement, they need to show editing with the gal-nac. Gal-nac conjugation delivers oNS via a different mechanism than LNPs and gal-nac conjugation results

in lower ON concentrations in hepatocytes than LNP. I could get into many other differences, but to use LNPs when gal-nacs are available is a giant leap backwards.

RESPONSE: In principle, I agree with the reviewer. In the revision we also show GalNAc-mediated uptake into primary hepatocyte and we now clearly state that GalNAc-mediated delivery is an important milestone for future drug development to achieve (lines 511 ff). Today, we are aware of two companies that have achieved this already with their clinical candidates for their AATD programs, Wave Life Sciences with WVE-006, and AIRNA with AIRNA-001. However, today, no details on the exact chemical compositions of the respective RNA drug candidates have been published in a peer-reviewed journal. Two other companies, KorroBio and likely also ProQR seem to implement rather LNP-based strategies. For KorroBio's clinical AATD program, the phase 1/2a trial NCT06677307 (<https://clinicaltrials.gov/study/NCT06677307>) has just started in which KRRO-110 is delivered intravenously after LNP formulation. For ProQR there is currently no clinical study in the RNA base editing field findable. A recent highlight of the landscape and clinical trials can be found here: <https://www.nature.com/articles/d41573-024-00070-y>.

Overall, I do not see that we would negatively influence the field. What we aim to publish is at the cutting edge where industry and academia are currently standing.

4. The mechanistic insights I would like to see are those that would DIRECT future SAR work. I am pleased that authors are evaluating protein interactions, but they offer no insights of value. (By the way, it has been clearly shown that the gapmer structure itself is not a major factor in protein binding or the formation of aggregates. Rather the lipophilicity of 2' modifications and the PS number and placement are the major factors).

RESPONSE: Well, we are aware that the overall hydrophobicity of the molecule is determining its toxicity, in particular regarding the interaction, aggregation, mislocalization and degradation with/of proteins. This said, RESTORE oligos differ clearly from gapmers and the initial data we provide in the revision show that RESTORE oligos lean less toward aggregate formation, but have reasonable interaction profiles with (nuclear) proteins and do not degrade NONO. Thus, I do not follow the concern expressed by the reviewer. Besides that, I wish to mention that our manuscript offers quite some insight into the general design rules, e.g. the clear nuclear mechanism of action, the interplay with the splicing process, the immune suppressive effect of chemical modification of the pyrimidines, the protein interactome, the preferred ASO (a)symmetry, the preferred ASO modification pattern around the central base triplet, etc. Thus, our insights are valuable to drug designers in industry and academia.

5. With high affinity modifications like 2'F and LNA, traditional folding programs are useless, but the authors could easily map the structure and should because the structure may provide insights into mechanism and challenges ahead.

RESPONSE: We are aware that certain high affinity modifications influence the folding of an oligonucleotide and that this is not reflected in typical web-based folding tools. However, our simple analysis would mark up if a certain sequence has a high degree of self-complementarity. This, in the past, was the reason for some guide RNAs not to function properly and thus we test for self-complementarity regularly. As the issue was brought up by one of the reviewers, not reviewer #2, we included the given, short analysis in the revision.

6. Finally, irrespective of the type of institution in which work is performed, I have to believe that the authors would agree that the conclusions and claims made in a manuscript must be fully supported by the data.

RESPONSE: We fully agree that any claim needs sufficient backing by data. However, we believe that this is given in our manuscript. We wish to mention that we have toned down further and added more clarifying and disclaiming language to avoid any overinterpretation regarding toxicity in the in vivo experiment.

4 (Remarks to the Author):

We thank the authors for the detailed and well-articulated responses to the reviewers' comments. We believe the changes improve an already good work.

I would also apologise for the poor wording of the first comment: the comment did not imply that the work was not interesting, it only meant to highlight the fact that what had made RESTORE stand out among similar linear ONs, like AIMers, was the inclusion of the ADAR recruitment domain and that removal of the domain strips away this distinguishing feature. As such, it might make sense to come up with a new name rather than call it RESTORE 2.0. It also does not sound fair to the other approaches the attempt to name a somehow identical strategy (despite all development behind it) as an update of a strategy that had fairly distinctive features.

RESPONSE: We thank the reviewer for this very positive feedback on our revision and rebuttal letter. A few words regarding the naming RESTORE 2.0. You are absolutely right that the ADAR recruitment motif is a unique feature in our designs to harness endogenous ADAR. We use it in the encodable CLUSTER guide RNA and used in the RESTORE 1.0 oligonucleotide approach. However, when you think about the repeated dosing of a chemically manufacturable RNA drug like RESTORE, you want to make the drug as short as possible. Thus, we kept the ADAR recruiting motif in the next generation of genetically encodable CLUSTER guide RNAs (see Reautschnig et al., Nature Biotech 2024), but for the RESTORE approach we aimed to make these oligos as short as possible. Our original idea was to only eliminate one half of the ADAR recruitment domain and to create a bulged "ADAR-recruiting motif" once the RESTORE oligo binds to the target mRNA. Consequently, we started our design process with ca. 60 nt oligos. However, we quickly found out that the oligos are more efficient if we shorten them down to 40 nt, given we do it in the right, asymmetric fashion. In 2022, Wave's AIMer platform was published and we continued to shorten the oligos further and found the unexpected sweet spot around 33 nt length. So, what seems less logic for you: for us it was a logical development over the last 6 years. Still, we would like to keep the name. The technology was licensed and transferred to our start-up company AIRNA where it was further developed. After quite some internal discussions, we decided to call the company pipeline RESTORE+, to emphasize the history and relation to our early work in the space. Today, AIRNA is sometimes seen as a fast follower even though it is actually a pioneering company. For this reason, we wish to stay with the name RESTORE 2.0 in the manuscript.

Minor comments:

- Reference 46 (Yi et al. Genome Biology (2023) 24:243) could be included in the introduction.
Lines 51-52

RESPONSE: You are right, we cite the reference now earlier and recite again later.

- Line 119 typo – written “from” instead of “form”

RESPONSE: The typo was corrected.

- Fig 3 – Text mentions N0 for orphan cytosine but figure shows C0.

RESPONSE: The figure was corrected.

- Line 363 pre-mRNA and mature mRNA comparison – maybe include a bit more detail about the target transcripts. Seems too vague the way it is.

RESPONSE: We included the new Supplementary Figure 8, to better explain the different stages of pre-mRNA splicing and how we assessed this by PCR.

- Line 486 should be “build” instead of “built”

RESPONSE: The typo was corrected.

- Line 506 “were” can be removed

RESPONSE: The typo was corrected.

Reviewer #5 (Remarks to the Author):

RESPONSE: Thank you for participating in the review process.

Rebuttal Letter

Reviewer #6 (Remarks to the Author):

The article by Pfeiffer et al. reports on the chemical optimization of short oligonucleotides (30–40 nt), demonstrating their efficiency in inducing robust editing both in vitro and in vivo (for a single target).

Developing fully chemically modified ADAR-recruiting oligonucleotides remains an area of significant research and commercial interest. Although such oligonucleotides have been published recently, achieving maximum potency typically requires the use of stereoselective PS and PN modifications, which are not yet widely available commercially.

The authors show that altering the 2'-O-methyl and 2'-fluoro modification pattern from a block design to a mixed design (i.e., a mixmer of 2'-O-methyl and 2'-fluoro residues) enhances editing efficiency in vitro. However, data is presented for only one concentration.

While the overall data quality is acceptable, the manuscript would need substantial rewriting to be suitable for a general audience.

RESPONSE: We thank the reviewer for this very positive feedback on our manuscript.

Major Comments

Outdated Context and Lack of Positioning Against Prior Work

The article appears to compile data accumulated over several years, presented from a historical perspective with additional content likely added in response to reviewer feedback. However, much of the experimental content is outdated given recent developments in the field. In particular, the work should be positioned clearly in the context of Monian et al., published in Nature Biotechnology (March 2022) [<https://www.nature.com/articles/s41587-022-01225-1>], which reported fully chemically modified short ADAR-targeting oligonucleotides.

Key Innovation Is Not Clearly Highlighted

Within this current landscape, the most significant contribution of the manuscript is not the development of fully chemically modified ADAR-targeting oligonucleotides (which has been done previously), but rather the demonstration that highly active ADAR-recruiting ASOs can be engineered without the need for stereoselective modifications. This makes the technology more broadly accessible to the research community. This is a valuable and important insight, but the message is currently diluted by the inclusion of extensive historical data and less relevant details.

RESPONSE: In the revision, we have now positioned the work much clearer in discussion against the state-of-the-art Aimer platform (Monian et al.), and attenuated the reference to our “historical” RESTORE 1.0 design. Along those lines, we specified the title (to “*Stereo-random Oligonucleotides Enable Efficient Recruitment of ADAR in vitro and in vivo*”), we rewrote the abstract and all other parts of the manuscript putting focus on how we achieve efficient editing on a stereo-random oligonucleotide design.

We also shortened the first two paragraphs “*Basic design principles of unstructured ADAR-recruiting ON*” and “*Partial stabilization by pyrimidine modification*”, which shows the older, somewhat “historical” data. However, we did not entirely remove this part from the manuscript for two reasons. First, we believe that some of the data is still valuable for the drug designer and also not yet sufficiently discussed in published work, e.g. effect of general PS content (Fig. 1C), symmetric versus asymmetric ON design (Fig. 1B, D), repetitive 2'-modifications (Fig. 1A,B), and how all this shapes the improved efficacy of the ON. Second, we wanted to include the discussion of data that was explicitly asked for by other reviewers, like data presented in Supplementary Figures 1, 2, 5-8, which are largely called out in the first half of the paper. However, we now always pick up the leitmotif again, positioning our data against the work of Monian et al., when we continue in the story line.

Use of Confusing Terminology ("Restore-1", "Restore-2")

The use of marketing-like terminology such as “Restore-1” and “Restore-2” is confusing and detracts from scientific clarity. While the intention to brand or classify the compounds is understandable, it adds little scientific value and may confuse general readers. It would be preferable to use clear, descriptive terms such as “fully chemically modified ADAR-recruiting oligonucleotides.” If the authors wish to introduce new terminology (e.g., “Restore”), it should be clearly defined in the discussion section as a proposed nomenclature for future reference. It’s worth noting that in the siRNA field, compounds are typically referred to by their modification profiles, unless part of a marketed or proprietary pipeline.

RESPONSE: In the revision, we now try to follow the recommendation of the reviewer and make less use of the term RESTORE 2.0 and limit its use for the final oligonucleotide designs that follow all our design rules and are fully modified.

Lack of Critical Evaluation of Study Limitations

The manuscript lacks a balanced discussion of the study’s limitations. For example, the claim that “Restore-2 is safe” is not adequately supported by the presented data. At best, the available data shows that in a single study, LNP-mediated delivery of the ADAR-recruiting ASO led to efficient editing three days post-treatment, without observable toxicity or animal mortality.

Regarding the safety of 2'-fluoro modifications: both the reviewers’ and authors’ points have merit, but the truth lies in the nuance. Toxicity of 2'-F-modified oligonucleotides in vivo depends on dose, modification pattern, and overall structure. In particular, concerns have been raised about protein binding and mitochondrial toxicity from high doses (~25 g/year, revusiran failed PHASE III) of fully modified siRNAs lacking 5'-terminal backbone stabilization. If the authors wish to support their claims, this would require in vivo evaluation of degradation kinetics using LC-MS (and I can confidently predict significantly lower stability compared to fully modified siRNAs), as well as analysis of clearance metabolites and exaggerated pharmacology—areas that fall outside the scope of this report.

While the oligonucleotides designed here appear sufficiently safe for research use—which is an important contribution—the current data does not inform on preclinical or clinical safety. Moreover, as single-stranded oligonucleotides, these ADAR-recruiting ASOs are (1) subject to known sequence-specific ASO toxicities, and (2) significantly less stable than duplex siRNAs, potentially resulting in different toxicity profiles upon repeated dosing.

None of this detracts from the scientific importance of the findings—but it does warrant a more objective presentation of the data and its implications.

RESPONSE: To avoid overinterpretation regarding the meaning of our data for preclinical and clinical application and safety, we have now removed any claims into that direction and any language around therapeutic use from the main text including the discussion. Additionally, we have now added a new *“Limitations of the study”* paragraph right after the discussion paragraph, where we start with your original statement: *“While the oligonucleotides designed here appear sufficiently safe for research use, the current data does not inform on preclinical or clinical safety”*. This is followed by a more general discussion of future engineering challenges towards improving efficiency and safety and the recommendation to lower the PS and 2'-F content in future designs. We hope that this satisfies your concerns and give a balanced and critical view on the impact of the study.

Rebuttal Letter

Reviewers' comments:

Reviewer #7 (Remarks to the Author):

The authors have sufficiently addressed the comments of Reviewer 6. These include references and comparisons to current state of the art AIMer technology, clear focus on the key innovation of the study, and a thorough evaluation of the limitations of the study. As such, I believe this manuscript is appropriate for publication.

One note. In lines 520-521 the authors say that analyzing the protein interactome will be helpful to understand the protein interactome. Please rephrase.

RESPONSE: The indicated sentence was rephrased to improve readability.